# PROVABLE MORE DATA HURT IN HIGH DIMENSIONAL LEAST SQUARES ESTIMATOR

## ABSTRACT

This paper investigates the finite-sample prediction risk of the high-dimensional least squares estimator. We derive the central limit theorem for the prediction risk when both the sample size and the number of features tend to infinity. Furthermore, the finite-sample distribution and the confidence interval of the prediction risk are provided. Our theoretical results demonstrate the sample-wise non-monotonicity of the prediction risk and confirm "more data hurt" phenomenon.

## 1 INTRODUCTION

More data hurt refers to the phenomenon that training on more data can hurt the prediction performance of the learned model, especially for some deep learning tasks. Loog et al. (2019) shows that various standard learners can lead to sample-wise non-monotonicity. Nakkiran et al. (2019) experimentally confirms the sample-wise non-monotonicity of the test accuracy on deep neural networks. This challenges the conventional understanding in large sample properties: if an estimator is consistent, more data makes the estimator more stable and improves its finite-sample performance. Nakkiran (2019) considers adding one single data point to a linear regression task and analyzes its marginal effect to the test risk. Dereziński et al. (2019) gives an exact non-asymptotic risk of the high-dimensional least squares estimator and observes the sample-wise non-monotonicity on mean square error. For adversarially robust models, Min et al. (2020) proves that more data may increase the gap between the generalization error of adversarially-trained models and standard models. Chen et al. (2020) shows that more training data causes the generalization error to increase in the strong adversary regime. In this work, we derive the finite-sample distribution of the prediction risk under linear models and prove the *"more data hurt"* phenomenon from an asymptotic point of view.

Intuitively, the *"more data hurt"* stems from the *"double descent"* risk curve: as the model complexity increases, the prediction risk of the learned model first decreases and then increases, and then decreases again. The *double descent* phenomenon can be precisely quantified for certain simple models (Hastie et al. (2019); Mei & Montanari (2019); Ba et al. (2019); Belkin et al. (2019); Bartlett et al. (2020); Xing et al. (2019)). Among these works, Hastie et al. (2019) and Mei & Montanari (2019) use the tools from random matrix theory and explicitly prove the double descent curve of the asymptotic risk of linear regression and random features regression in high dimensional setup. Ba et al. (2019) gives the asymptotic risk of two-layer neural networks when either the first or the second layer is trained using a gradient flow.

The second decline of the prediction risk in the double descent curve is highly related to the *more data hurt* phenomenon. In the over-parameterized regime when the model complexity is fixed while the sample size increases, the degree of over-parameterization decreases and becomes close to the interpolation boundary (for example $p/n = 1$ in Hastie et al. (2019)), in which a high prediction risk is achieved. However, the existing asymptotic results, which focus on the first-order limit of the prediction risk, cannot fully describe the *more data hurt* phenomenon. In fact, the *"double descent"* curve is a function of the limiting ratio $\lim p/n$, which may not be able to characterize the empirical prediction risk in finite sample situations. There will be a non-negligible discrepancy between the empirical prediction risk and its limit, especially when the sample size or dimension is small. Fine-grained second-order results are thus needed to fully characterize such discrepancy and further, a confidence band for the prediction risk can be constructed to evaluate its finite sample performance. We take Figure 1 as an example to illustrate this. According to the first-order limit, given a fixed dimension $p = 100$, the prediction risks at sample size $n = 90$ and $n = 98$ are about 10.20 and

49.02. More data hurt seems true. However, the 95% confidence interval of the prediction risks with sample size 98 is $[4.91, 142.12]$, which contains the risk for $n = 90$. Then more data hurt is not statistically significant. Hence, in this work, we characterize the second-order fluctuations of the prediction risk and make attempts to fill this gap. We employ the linear regression task in Hastie et al. (2019) and Nakkiran (2019), and introduce new tools from the random matrix theory, e.g. the central limit theorems for linear spectral statistics in Bai & Silverstein (2004); Bai et al. (2007), to derive the central limit theorem of the prediction risk.

Consider a linear regression task with $n$ data points and $p$ features, the setup of the *more data hurt* is similar to that in the classical asymptotic analysis in Van der Vaart (2000). According to the classical asymptotic analysis with $p$ fixed and $n \to \infty$, the least square estimator is unbiased and $\sqrt{n}$-consistent to the ground truth. This implies that the more data will not hurt and even improve the prediction performance. However, the story is very different in the over-parameterized regime. The prediction risk doesn't decrease monotonously with $n$ when $p > n$. More data does hurt in the over-parameterized case. In the following, we will justify this phenomenon by developing the second-order asymptotic results as both $n$ and $p$ tend to infinity. We assume $p/n \to c$, and denote $0 < n_1 < n_2 < +\infty$, $c_1 = p/n_1$ and $c_2 = p/n_2$. Then the direct comparison of the prediction risk between sample sizes $n_1$ and $n_2$ can be decomposed into three parts: (i) the gap between the finite-sample risk under $n = n_1$ and the asymptotic risk with $c = c_1$; (ii) the gap between the finite-sample risk under $n = n_2$ and the asymptotic risk with $c = c_2$; (iii) the comparison between two asymptotic risks under $c = c_1$ and $c = c_2$. Theorem 1 and 2 of Hastie et al. (2019) give answers to the task (iii). For (i) and (ii), we develop in this paper the convergence rate and the limiting distribution of the prediction risk as $n, p \to +\infty$, $p/n \to c$. Furthermore, the confidence interval of the finite-sample risk can be obtained as well.

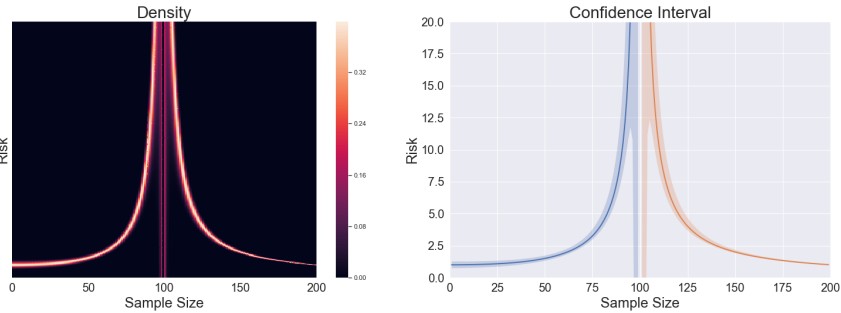

Figure 1: Sample-wise double descent. We take $p = 100$ and $1 \le n \le 200$. **Left**: The conditional density of the prediction risk when sample size varies from 1 to 200. According to the conditional distribution of the prediction risk, we can observe the sample-wise double descent phenomenon. **Right**: The 95%-confidence band (point-wise) of the prediction risk. In the over-parameterized regime $1 \le n < 100$, there exists some pairs $(n_1, n_2)$, $1 \le n_1 < n_2 < 100$ such that the upper boundary of the confidence interval at $n_1$ is smaller than the lower boundary of the confidence interval at $n_2$. This confirms the more data hurt phenomenon.

The main goal of this paper is to study the second order asymptotic behavior of two different types of conditional prediction risk in the linear regression model. One is $R_{\mathbf{X}, \boldsymbol{\beta}}(\hat{\boldsymbol{\beta}}, \boldsymbol{\beta})$ given both the training data and regression coefficient while the other is $R_{\mathbf{X}}(\hat{\boldsymbol{\beta}}, \boldsymbol{\beta})$ given the training data only. We summarize our main results as follows: (1) The regression coefficient is set to be either random or nonrandom to cover more cases. Different convergence rates and limiting distributions of both prediction risk are derived under various scenarios. (2) In particular, the finite-sample distribution of the conditional prediction risk given both the training data and regression coefficient is derived and the sample-wise double descent is characterized in Theorem 4.2 and Theorem 4.5 (see Figure 1). Under certain assumptions, the more data hurt phenomenon can be confirmed by comparing the confidence intervals built via the central limit theorems. (3) Our results incorporate non-Gaussian observations. For Gaussian data, the limiting mean and variance in the central limit theorems have simpler forms, see Section 4.2 and 4.3 for more details.

The rest of this paper is organized as follows. Section 3 introduces the model settings and two different prediction risk. Section 4 presents the main results on CLTs for the two types of risk with

discussion. Section 5 conducts simulation experiments to verify the main results. All the technical proofs and lemmas are relegated to the appendix in the supplementary file.

## 2 RELATED WORK

**Double Descent** The *double descent* curve describes how generalization ability changes as model capacity increases. It subsumes the classical bias-variance trade-off, a U-shape curve, and further show that the test error exhibits a second drop when the model capacity exceeds the interpolation threshold (Belkin et al. (2018); Geiger et al. (2019); Spigler et al. (2019); Advani & Saxe (2017)). The *double descent* phenomenon has been quantified for certain models, including two-layer neural networks via non-asymptotic bounds or asymptotic risk (Belkin et al. (2019); Muthukumar et al. (2020); Hastie et al. (2019); Mei & Montanari (2019); Ba et al. (2019)). As our results are based on the linear regression, we mainly focus on the literature of linear models. Muthukumar et al. (2020) and Bartlett et al. (2020) derive the generalization bounds for over-parametrized linear models and show the benefits of the interpolation. Hastie et al. (2019) gives the first-order limit of the generalization error for linear regressions as $n, p \to +\infty$. Dereziński et al. (2019) provides an exact non-asymptotic expression for *double descent* of the high-dimensional least square estimator. Wu & Xu (2020) extends the first-order limit of the prediction error of the generalized weighted ridge estimator to more general case with anisotropic features and signals. Montanari et al. (2019), Deng et al. (2019) and Kini & Thrampoulidis (2020) investigate the sharp asymptotic of binary classification tasks with the max-margin solution and the maximum likelihood solution. Emami et al. (2020) and Gerbelot et al. (2020a) consider the *double descent* in generalized linear models. Furthermore, the *double descent* phenomenon is also observed on linear tasks with various problems and assumptions, e.g. LeJeune et al. (2020); Gerbelot et al. (2020b); Javanmard et al. (2020); Dar & Baraniuk (2020); Xu & Hsu (2019); Dar et al. (2020). Xing et al. (2019) sharply quantifies the benefit of interpolation in the nearest neighbors algorithm. Mei & Montanari (2019) derives the limiting risk on the random features model and shows that minimum generalization error is achieved by highly over-parametrized interpolators. Ba et al. (2019) gives the limiting risk of the regression problem under two-layer neural networks. However, the existing asymptotic results focus on the first-order limit of the prediction risk and do not indicate the convergence rate. There are very few second-order results in the literature, Shen & Bellec (2020) establishes the asymptotic normality for the derivatives of 2-layers neural network, but not the exact limiting distribution of the risk. In this work, we are the first to develop results on second-order fluctuations of the prediction risk in linear regressions and provide its corresponding confidence intervals. The more data hurt phenomenon is further justified from the asymptotic point of view.

**Random Matrix Theory** The primary tool for analyzing the second-order fluctuations of prediction risk comes from random matrix theory. In particular, Bai & Silverstein (2004) refines the central limit theorem for linear spectral statistics of large dimensional sample covariance matrix with general population and the population is not necessary to be Gaussian. Similar central limit theorems are also developed for other random matrix ensembles, see Sinai & Soshnikov (1998); Bai & Yao (2005); Zheng (2012). Other than the central limit theorem for linear spectral statistics, Bai et al. (2007) and Pan & Zhou (2008) study the asymptotic fluctuation of eigenvectors of sample covariance matrices. Bai & Yao (2008) considers the fluctuation of quadratic forms. All these technical tools and results are adopted and fully utilized in this paper, especially those related to Stieltjes transform, which are closely connected to the prediction risk studied in this paper.

## 3 PRELIMINARIES

### 3.1 PROBLEM, DATA AND ESTIMATOR

Suppose that the training data $\{(\mathbf{x}_i, \mathbf{y}_i) \in \mathbb{R}^p \times \mathbb{R}, i = 1, 2, \dots, n\}$ is generated independently from the model (ground truth or teacher model):

$$\mathbf{y}_i = \boldsymbol{\beta}^\mathrm{T} \mathbf{x}_i + \epsilon_i, \quad \text{and} \quad (\mathbf{x}_i, \epsilon_i) \sim (P_\mathbf{x}, P_\epsilon), \quad i = 1, 2, \dots, n. \tag{1}$$

Here, $P_\mathbf{x}$ is a distribution on $\mathbb{R}^p$ such that $\mathbb{E}(\mathbf{x}_i) = \mathbf{0}$, $\mathrm{Cov}(\mathbf{x}_i) = \boldsymbol{\Sigma}$, and $P_\epsilon$ is a distribution on $\mathbb{R}$ such that $\mathbb{E}(\epsilon_i) = 0$, $\mathrm{Var}(\epsilon_i) = \sigma^2$. In particular, the coordinates of $\mathbf{x}_i$ are not necessarily

independent, that is, $\boldsymbol{\Sigma}$ is not restricted to be diagonal. To proceed further, we denote

$$\mathbf{X}_{n\times p} = (\mathbf{x}_1, \mathbf{x}_2, \ldots, \mathbf{x}_n)^{\mathrm{T}}, \quad \mathbf{y} = (\mathrm{y}_1, \mathrm{y}_2, \ldots, \mathrm{y}_n)^{\mathrm{T}}.$$

The minimum $\ell_2$ norm (min-norm) least squares estimator, of $\mathbf{y}$ on $\mathbf{X}$, is defined by

$$\hat{\boldsymbol{\beta}} = \arg\min_{\boldsymbol{\beta}} \|\mathbf{y} - \mathbf{X}\boldsymbol{\beta}\|^2 = (\mathbf{X}^{\mathrm{T}}\mathbf{X})^+ \mathbf{X}^{\mathrm{T}}\mathbf{y}, \tag{2}$$

where $(\mathbf{X}^{\mathrm{T}}\mathbf{X})^+$ denotes the Moore-Penrose pseudoinverse of $\mathbf{X}^{\mathrm{T}}\mathbf{X}$.

## 3.2 BIAS, VARIANCE AND RISK

Similar to Hastie et al. (2019), we define two different types of out-of-sample prediction risk. The first one is given by

$$R_{\mathbf{X}}(\hat{\boldsymbol{\beta}}, \boldsymbol{\beta}) = \mathbb{E}\big[(\mathbf{x}_0^{\mathrm{T}}\hat{\boldsymbol{\beta}} - \mathbf{x}_0^{\mathrm{T}}\boldsymbol{\beta})^2\big|\mathbf{X}\big] = \mathbb{E}\big[\|\hat{\boldsymbol{\beta}} - \boldsymbol{\beta}\|_{\boldsymbol{\Sigma}}^2\big|\mathbf{X}\big], \tag{3}$$

where $\mathbf{x}_0 \sim P_{\mathbf{x}}$ is a test point and is independent of the training data, and the notation $\|\boldsymbol{\beta}\|_{\boldsymbol{\Sigma}}^2$ stands for $\boldsymbol{\beta}^{\mathrm{T}}\boldsymbol{\Sigma}\boldsymbol{\beta}$. Here $\boldsymbol{\beta}$ is assumed to be a random vector independent of $\mathbf{x}_0$. In this definition, the expectation $\mathbb{E}$ stands for the conditional expectation for $\mathbf{x}_0$, $\hat{\boldsymbol{\beta}}$ and $\boldsymbol{\beta}$ when $\mathbf{X}$ is given. According to the bias-variance decomposition, we have $R_{\mathbf{X}}(\hat{\boldsymbol{\beta}}, \boldsymbol{\beta}) := B_{\mathbf{X}}(\hat{\boldsymbol{\beta}}, \boldsymbol{\beta}) + V_{\mathbf{X}}(\hat{\boldsymbol{\beta}}, \boldsymbol{\beta})$, where

$$B_{\mathbf{X}}(\hat{\boldsymbol{\beta}}, \boldsymbol{\beta}) = \mathbb{E}\Big\{\|\mathbb{E}(\hat{\boldsymbol{\beta}}|\mathbf{X}) - \boldsymbol{\beta}\|_{\boldsymbol{\Sigma}}^2\big|\mathbf{X}\Big\} \quad \text{and} \quad V_{\mathbf{X}}(\hat{\boldsymbol{\beta}}, \boldsymbol{\beta}) = \mathrm{Tr}\{\mathrm{Cov}(\hat{\boldsymbol{\beta}}|\mathbf{X})\boldsymbol{\Sigma}\}. \tag{4}$$

Plugging the model (1) into the min-norm estimator (2), the bias and variance terms can be rewritten as

$$B_{\mathbf{X}}(\hat{\boldsymbol{\beta}}, \boldsymbol{\beta}) = \mathbb{E}\big\{\boldsymbol{\beta}^{\mathrm{T}}\boldsymbol{\Pi}\boldsymbol{\Sigma}\boldsymbol{\Pi}\boldsymbol{\beta}\big|\mathbf{X}\big\} \quad \text{and} \quad V_{\mathbf{X}}(\hat{\boldsymbol{\beta}}, \boldsymbol{\beta}) = \frac{\sigma^2}{n}\mathrm{Tr}(\hat{\boldsymbol{\Sigma}}^+\boldsymbol{\Sigma}),$$

where $\hat{\boldsymbol{\Sigma}} = \mathbf{X}^{\mathrm{T}}\mathbf{X}/n$ is the (uncentered) sample covariance matrix of $\mathbf{X}$, and $\boldsymbol{\Pi} = \boldsymbol{I}_p - \hat{\boldsymbol{\Sigma}}^+\hat{\boldsymbol{\Sigma}}$ is the projection onto the null space of $\mathbf{X}$.

The second type of out-of-sample prediction risk is defined as

$$R_{\mathbf{X},\boldsymbol{\beta}}(\hat{\boldsymbol{\beta}}, \boldsymbol{\beta}) = \mathbb{E}\big[(\mathbf{x}_0^{\mathrm{T}}\hat{\boldsymbol{\beta}} - \mathbf{x}_0^{\mathrm{T}}\boldsymbol{\beta})^2\big|\mathbf{X}, \boldsymbol{\beta}\big] = \mathbb{E}\big[\|\hat{\boldsymbol{\beta}} - \boldsymbol{\beta}\|_{\boldsymbol{\Sigma}}^2\big|\mathbf{X}, \boldsymbol{\beta}\big], \tag{5}$$

where

$$B_{\mathbf{X},\boldsymbol{\beta}}(\hat{\boldsymbol{\beta}}, \boldsymbol{\beta}) = \boldsymbol{\beta}^{\mathrm{T}}\boldsymbol{\Pi}\boldsymbol{\Sigma}\boldsymbol{\Pi}\boldsymbol{\beta} \quad \text{and} \quad V_{\mathbf{X},\boldsymbol{\beta}}(\hat{\boldsymbol{\beta}}, \boldsymbol{\beta}) = V_{\mathbf{X}}(\hat{\boldsymbol{\beta}}, \boldsymbol{\beta}) = \frac{\sigma^2}{n}\mathrm{Tr}(\hat{\boldsymbol{\Sigma}}^+\boldsymbol{\Sigma}).$$

In this definition, the parameter $\boldsymbol{\beta}$ is assumed to be given. The expectation $\mathbb{E}$ is the conditional expectation for $\mathbf{x}_0$ and $\hat{\boldsymbol{\beta}}$ when $\mathbf{X}$ and $\boldsymbol{\beta}$ are given. This is consistent with the commonly-used testing procedure, in which a trained model is evaluated by the average loss on unseen testing data.

## 4 MAIN RESULTS

Before stating our main results, we briefly highlight the challenges we faced in proving the *more data hurt* phenomenon. First, the finite-sample behaviors of the prediction risk is required. Hastie et al. (2019) gives the first-order limits of both $R_{\mathbf{X},\boldsymbol{\beta}}(\hat{\boldsymbol{\beta}}, \boldsymbol{\beta})$ and $R_{\mathbf{X}}(\hat{\boldsymbol{\beta}}, \boldsymbol{\beta})$ as $n, p \to +\infty$ and $p/n \to c \in (0, +\infty)$. However, to prove the *more data hurt* phenomenon, we should fix $p$ and investigate the finite-sample risk with sample size $n$ varies. This implies that only knowing the first-order limit is not enough, the convergence rate is also needed. To solve this problem, we have derived the central limit theorems for both $R_{\mathbf{X},\boldsymbol{\beta}}(\hat{\boldsymbol{\beta}}, \boldsymbol{\beta})$ and $R_{\mathbf{X}}(\hat{\boldsymbol{\beta}}, \boldsymbol{\beta})$, respectively, which characterize the second-order fluctuations of the risk. Then we can figure out the finite-sample behavior of the risk by computing the gap between the risk and its limit. The confidence intervals of the risk can be further obtained. Second, the parameter $\boldsymbol{\beta}$ also contributes randomness to the finite-sample risk, which further influences the convergence rate. To analyze the contribution of $\boldsymbol{\beta}$, we need to make use of the technical tools and asymptotic results for eigenvectors and quadratic forms developed in Bai et al. (2007) and Bai & Yao (2008). Another interesting finding is that, in the over-parameterized regime such that $p > n$, the two types of out-of-sample prediction risk $R_{\mathbf{X},\boldsymbol{\beta}}(\hat{\boldsymbol{\beta}}, \boldsymbol{\beta})$ and $R_{\mathbf{X}}(\hat{\boldsymbol{\beta}}, \boldsymbol{\beta})$ enjoy different convergence rates.

### 4.1 ASSUMPTIONS AND MORE NOTATIONS

As follows are some notations used in this paper. The $p \times p$ identity matrix is denoted by $\boldsymbol{I}_p$. For a symmetric matrix $\boldsymbol{A} \in \mathbb{R}^{p \times p}$, we define its empirical spectral distribution as

$$F^{\boldsymbol{A}}(x) = \frac{1}{p} \sum_{i=1}^{p} \mathbb{1}\{\lambda_i(\boldsymbol{A}) \leq x\}$$

where $\mathbb{1}\{\cdot\}$ is the indicator function and $\lambda_i(\boldsymbol{A})$, $i = 1, 2, \ldots p$ are the eigenvalues of $\boldsymbol{A}$. The notation $\xrightarrow{d}$ stands for the convergence in distribution. $Z_{\alpha/2}$ is the $\alpha/2$ upper quantile of the standard normal distribution, $\lambda_{\max}(\boldsymbol{A})$ and $\lambda_{\min}(\boldsymbol{A})$ denote the largest and smallest eigenvalues of $\boldsymbol{A}$, respectively.

Here we list all the assumptions for $\mathbf{X}$ and $\boldsymbol{\beta}$ needed under different scenarios:

(A) $\mathbf{x}_j \sim P_{\mathbf{x}}$ is of the form $\mathbf{x}_j = \boldsymbol{\Sigma}^{1/2}\mathbf{z}_j$, where $\mathbf{z}_j$ is a $p$-length random vector with i.i.d. entries that have zero mean, unit variance, and a finite 4-th order moment $\mathbb{E}(\mathbf{z}_{ij}^4) = \nu_4$, $i = 1, \cdots, p, j = 1, \cdots, n$.

(B1) $\boldsymbol{\Sigma}$ is a deterministic positive definite matrix, such that $0 < c_0 \leq \lambda_{\min}(\boldsymbol{\Sigma}) \leq \lambda_{\max}(\boldsymbol{\Sigma}) \leq c_1$, for all $n, p$ and some constants $c_0, c_1$. As $p \to \infty$, we assume that the empirical spectral distribution $F^{\boldsymbol{\Sigma}}$ converges weakly to a probability measure $H$.

(B2) $\boldsymbol{\Sigma}$ is an identity matrix, $\boldsymbol{\Sigma} = \boldsymbol{I}_p$.

(C1) $\boldsymbol{\beta}$ is a nonrandom constant vector, and $\|\boldsymbol{\beta}\|_2^2 = \boldsymbol{\beta}^{\mathrm{T}}\boldsymbol{\beta} = r^2$.

(C2) $\boldsymbol{\beta} \sim P_{\boldsymbol{\beta}}$ is independent of $\mathbf{X}$ and follows multivariate Gaussian distribution $\mathcal{N}_p(\mathbf{0}, \frac{r^2}{p}\boldsymbol{I}_p)$.

Throughout this paper, we consider the limiting distributions and the convergence rates of the out-of- sample prediction risk when $n, p \to \infty$ such that $p/n = c_n \to c \in (0, \infty)$. If $c > 1$, the sample size $n$ is smaller than the number of parameters $p$, we call this case *"over-parametrized"*. Otherwise when $c < 1$, we call it *"under-parameterized"*.

### 4.2 UNDER-PARAMETRIZED ASYMPTOTICS

In this section, we focus on the risk of the min-norm estimator (2) in the under-parametrized regime. According to Theorem 1 of Hastie et al. (2019), both $B_{\mathbf{X},\boldsymbol{\beta}}(\hat{\boldsymbol{\beta}}, \boldsymbol{\beta})$ and $B_{\mathbf{X}}(\hat{\boldsymbol{\beta}}, \boldsymbol{\beta})$ converge to $\sigma^2 c/(1-c)$ almost surely. The following Theorem 4.1 and 4.2 show that both $B_{\mathbf{X}}(\hat{\boldsymbol{\beta}}, \boldsymbol{\beta})$ and $B_{\mathbf{X},\boldsymbol{\beta}}(\hat{\boldsymbol{\beta}}, \boldsymbol{\beta})$ converge to $\sigma^2 c/(1-c)$ at the rate $1/p$. Furthermore, the limiting distributions are derived by making use of the CLT for linear spectral statistics of large-dimensional sample covariance matrices.

**Theorem 4.1.** *Suppose that the training data is generated from the model* (1)*, and the assumptions* (A) *and* (B1) *hold. Then the first type of out-of-sample prediction risk* $R_{\mathbf{X}}(\hat{\boldsymbol{\beta}}, \boldsymbol{\beta})$ *of the min-norm estimator* (2) *satisfies that, as* $n, p \to \infty$ *such that* $p/n = c_n \to c < 1$,

$$p\left(R_{\mathbf{X}}(\hat{\boldsymbol{\beta}}, \boldsymbol{\beta}) - \frac{c_n\sigma^2}{1-c_n}\right) \xrightarrow{d} N(\mu_c, \sigma_c^2), \tag{6}$$

*where*

$$\mu_c = \frac{c^2\sigma^2}{(c-1)^2} + \frac{\sigma^2 c^2(\nu_4 - 3)}{1-c} \quad and \quad \sigma_c^2 = \frac{2c^3\sigma^4}{(c-1)^4} + \frac{c^3\sigma^4(\nu_4 - 3)}{(1-c)^2}.$$

*Conclusively,*

$$P(L_{\alpha,c} \leq R_{\mathbf{X}}(\hat{\boldsymbol{\beta}}, \boldsymbol{\beta}) \leq U_{\alpha,c}) \to 1 - \alpha, \tag{7}$$

*where* $1 - \alpha$ *is the confidence level and*

$$L_{\alpha,c} = \frac{c_n\sigma^2}{1-c_n} + \frac{1}{p}(\mu_c - Z_{\alpha/2}\sigma_c), \quad U_{\alpha,c} = \frac{c_n\sigma^2}{1-c_n} + \frac{1}{p}(\mu_c + Z_{\alpha/2}\sigma_c).$$

Under the assumptions of Theorem 4.1, we know that $\boldsymbol{\Pi} = \boldsymbol{I}_p - \hat{\boldsymbol{\Sigma}}^+\hat{\boldsymbol{\Sigma}} = \mathbf{0}$ and

$$B_{\mathbf{X}}(\hat{\boldsymbol{\beta}}, \boldsymbol{\beta}) = B_{\mathbf{X},\boldsymbol{\beta}}(\hat{\boldsymbol{\beta}}, \boldsymbol{\beta}) = 0, \quad V_{\mathbf{X}}(\hat{\boldsymbol{\beta}}, \boldsymbol{\beta}) = V_{\mathbf{X},\boldsymbol{\beta}}(\hat{\boldsymbol{\beta}}, \boldsymbol{\beta}) = \frac{\sigma^2}{n}\mathrm{Tr}(\hat{\boldsymbol{\Sigma}}^+\boldsymbol{\Sigma}).$$

Thus $R_{\mathbf{X}}(\hat{\boldsymbol{\beta}}, \boldsymbol{\beta})$ equals to $R_{\mathbf{X},\boldsymbol{\beta}}(\hat{\boldsymbol{\beta}}, \boldsymbol{\beta})$ and the two risk share the same asymptotic limit.

**Theorem 4.2.** *Under the assumptions of Theorem 4.1, the second type of out-of-sample prediction risk $R_{\mathbf{X},\boldsymbol{\beta}}(\hat{\boldsymbol{\beta}},\boldsymbol{\beta})$ of the min-norm estimator (2) satisfies that, as $n,p \to \infty$ such that $p/n = c_n \to c < 1$,*

$$p\big(R_{\mathbf{X},\boldsymbol{\beta}}(\hat{\boldsymbol{\beta}},\boldsymbol{\beta}) - \frac{c_n\sigma^2}{1-c_n}\big) \xrightarrow{d} N(\mu_c, \sigma_c^2),$$

*and*

$$P(L_{\alpha,c} \leq R_{\mathbf{X},\boldsymbol{\beta}}(\hat{\boldsymbol{\beta}},\boldsymbol{\beta}) \leq U_{\alpha,c}) \to 1 - \alpha,$$

*where $\mu_c$, $\sigma_c^2$, $L_{\alpha,c}$ and $U_{\alpha,c}$ are the same as those in Theorem 4.1.*

### 4.3 OVER-PARAMETRIZED ASYMPTOTICS

In this section, we consider the min-norm estimator (2) in the over-parametrized case $c > 1$. The bias term, either $B_{\mathbf{X}}(\hat{\boldsymbol{\beta}},\boldsymbol{\beta})$ or $B_{\mathbf{X},\boldsymbol{\beta}}(\hat{\boldsymbol{\beta}},\boldsymbol{\beta})$, is generally nonzero. According to Lemma 2 in Hastie et al. (2019), both $B_{\mathbf{X}}(\hat{\boldsymbol{\beta}},\boldsymbol{\beta})$ and $B_{\mathbf{X},\boldsymbol{\beta}}(\hat{\boldsymbol{\beta}},\boldsymbol{\beta})$ converge to $r^2(1 - 1/c)$ as $n,p \to +\infty$ and $p/n \to c > 1$. This implies that the bias term can influence the asymptotic behavior of the prediction risk, including the convergence rate. Hence to derive the CLT of the out-of-sample prediction risk, we need to consider both the bias and variance terms in (4).

In the following, we investigate the asymptotic properties of the two prediction risk $R_{\mathbf{X}}(\hat{\boldsymbol{\beta}},\boldsymbol{\beta})$ and $R_{\mathbf{X},\boldsymbol{\beta}}(\hat{\boldsymbol{\beta}},\boldsymbol{\beta})$ under various combinations of the assumptions (A1), (B2) for $\mathbf{X}$ and scenarios (C1), (C2) for both random and nonrandom $\boldsymbol{\beta}$. We start with the case when $\boldsymbol{\beta}$ is a constant vector.

**Theorem 4.3.** *Suppose that the training data is generated from the model (1), and the assumptions (A), (B2) and (C1) hold. Then the first type of out-of-sample prediction risk $R_{\mathbf{X}}(\hat{\boldsymbol{\beta}},\boldsymbol{\beta})$ of the min-norm estimator (2) satisfies that, as $n,p \to \infty$ such that $p/n = c_n \to c > 1$,*

$$\sqrt{p}\Big\{R_{\mathbf{X}}(\hat{\boldsymbol{\beta}},\boldsymbol{\beta}) - (1 - \frac{1}{c_n})r^2 - \frac{\sigma^2}{c_n - 1}\Big\} \xrightarrow{d} N(\mu_{c,1}, \sigma_{c,1}^2), \tag{8}$$

*where $\mu_{c,1} = 0$ and $\sigma_{c,1}^2 = \frac{2(c-1)}{c^2}r^4$. A more practical version is to replace $\mu_{c,1}$ and $\sigma_{c,1}^2$ with*

$$\tilde{\mu}_{c,1} = \frac{1}{\sqrt{p}}\Big\{\frac{c\sigma^2}{(1-c)^2} + \frac{\sigma^2(\nu_4 - 3)}{c-1}\Big\} \quad \text{and} \quad \tilde{\sigma}_{c,1}^2 = \frac{2(c-1)}{c^2}r^4 + \frac{1}{p}\Big\{\frac{2c^3\sigma^4}{(1-c)^4} + \frac{c\sigma^4(\nu_4 - 3)}{(c-1)^2}\Big\}.$$

*Conclusively,*

$$P(L_{\alpha,c} \leq R_{\mathbf{X}}(\hat{\boldsymbol{\beta}},\boldsymbol{\beta}) \leq U_{\alpha,c}) \to 1 - \alpha, \tag{9}$$

*where $1 - \alpha$ is the confidence level and*

$$L_{\alpha,c} = (1 - \frac{1}{c_n})r^2 + \frac{\sigma^2}{c_n - 1} + \frac{1}{\sqrt{p}}(\tilde{\mu}_{c,1} - Z_{\alpha/2}\tilde{\sigma}_{c,1}),$$

$$U_{\alpha,c} = (1 - \frac{1}{c_n})r^2 + \frac{\sigma^2}{c_n - 1} + \frac{1}{\sqrt{p}}(\tilde{\mu}_{c,1} + Z_{\alpha/2}\tilde{\sigma}_{c,1}).$$

**Remark 4.1.** *Under assumption (C1), $B_{\mathbf{X}}(\hat{\boldsymbol{\beta}},\boldsymbol{\beta}) = B_{\mathbf{X},\boldsymbol{\beta}}(\hat{\boldsymbol{\beta}},\boldsymbol{\beta})$ and $R_{\mathbf{X}}(\hat{\boldsymbol{\beta}},\boldsymbol{\beta}) = R_{\mathbf{X},\boldsymbol{\beta}}(\hat{\boldsymbol{\beta}},\boldsymbol{\beta})$. Thus Theorem 4.3 still holds if we replace $R_{\mathbf{X}}(\hat{\boldsymbol{\beta}},\boldsymbol{\beta})$ with $R_{\mathbf{X},\boldsymbol{\beta}}(\hat{\boldsymbol{\beta}},\boldsymbol{\beta})$.*

**Remark 4.2.** *Under Assumption (B2), the eigenvector of $\hat{\Sigma}$ is asymptotically Haar distributed. Therefore, the bias term $B_{\mathbf{X}}(\hat{\boldsymbol{\beta}},\boldsymbol{\beta})$ is only related to the length of $\boldsymbol{\beta}$. However, in the anisotropic settings with general $\Sigma$, the eigenvector of the $\hat{\Sigma}$ is no longer asymptotically Haar distributed. The limiting behavior of $B_{\mathbf{X}}(\hat{\boldsymbol{\beta}},\boldsymbol{\beta})$ heavily relies on the interaction between $\boldsymbol{\beta}$ and the eigenvectors of $\hat{\Sigma}$. Therefore, we conjecture that there is no universal convergence rate for the bias term $B_{\mathbf{X}}(\hat{\boldsymbol{\beta}},\boldsymbol{\beta})$ that can cover arbitrary non-random $\boldsymbol{\beta}$ and anisotropic $\Sigma$ in the over-parametrized case, not to mention the prediction risk $R_{\mathbf{X}}(\hat{\boldsymbol{\beta}},\boldsymbol{\beta})$. A small simulation experiment is conducted in Appendix G to confirm our conjecture on this point.*

Next we consider the case when $\boldsymbol{\beta}$ is a random vector that follows assumption (C2).

**Theorem 4.4.** *Suppose that the training data is generated from the model* (1)*, and the assumptions* (A)*,* (B2) *and* (C2) *hold. Then as* $n, p \to \infty$ *such that* $p/n = c_n \to c > 1$*, the first type of out-of-sample prediction risk* $R_{\mathbf{X}}(\hat{\boldsymbol{\beta}}, \boldsymbol{\beta})$ *of the min-norm estimator* (2) *satisfies,*

$$p\Big\{ R_{\mathbf{X}}(\hat{\boldsymbol{\beta}}, \boldsymbol{\beta}) - (1 - \frac{1}{c_n})r^2 - \frac{\sigma^2}{c_n - 1} \Big\} \xrightarrow{d} N(\mu_{c,2}, \sigma_{c,2}^2),$$

*where*

$$\mu_{c,2} = \frac{c\sigma^2}{(1-c)^2} + \frac{\sigma^2(\nu_4 - 3)}{c - 1} \quad and \quad \sigma_{c,2}^2 = \frac{2c^3\sigma^4}{(1-c)^4} + \frac{c\sigma^4(\nu_4 - 3)}{(c-1)^2}.$$

*Hence we have*

$$P(L_{\alpha,c} \leq R_{\mathbf{X}}(\hat{\boldsymbol{\beta}}, \boldsymbol{\beta}) \leq U_{\alpha,c}) \to 1 - \alpha,$$

*where*

$$L_{\alpha,c} = \frac{\sigma^2}{c_n - 1} + (1 - \frac{1}{c_n})r^2 + \frac{1}{p}(\mu_{c,2} - Z_{\alpha/2}\sigma_{c,2}),$$

$$U_{\alpha,c} = \frac{\sigma^2}{c_n - 1} + (1 - \frac{1}{c_n})r^2 + \frac{1}{p}(\mu_{c,2} + Z_{\alpha/2}\sigma_{c,2}).$$

As for $R_{\mathbf{X}, \boldsymbol{\beta}}(\hat{\boldsymbol{\beta}}, \boldsymbol{\beta})$, we have the following theorem.

**Theorem 4.5.** *Suppose that the training data is generated from the model* (1)*, and the assumptions* (A)*,* (B2) *and* (C2) *hold. Then, as* $n, p \to \infty$ *such that* $p/n = c_n \to c > 1$*, the second type of out-of-sample prediction risk* $R_{\mathbf{X}, \boldsymbol{\beta}}(\hat{\boldsymbol{\beta}}, \boldsymbol{\beta})$ *of the min-norm estimator* (2) *satisfies,*

$$\sqrt{p}\Big\{ R_{\mathbf{X}, \boldsymbol{\beta}}(\hat{\boldsymbol{\beta}}, \boldsymbol{\beta}) - (1 - \frac{1}{c_n})r^2 - \frac{\sigma^2}{c_n - 1} \Big\} \xrightarrow{d} N(\mu_{c,3}, \sigma_{c,3}^2), \tag{10}$$

*where* $\mu_{c,3} = 0$ *and* $\sigma_{c,3}^2 = 2(1 - \frac{1}{c})r^4$*. A more practical version is to replace* $\mu_{c,3}$ *and* $\sigma_{c,3}^2$ *with*

$$\tilde{\mu}_{c,3} = \frac{1}{\sqrt{p}}\Big\{ \frac{c\sigma^2}{(1-c)^2} + \frac{\sigma^2(\nu_4 - 3)}{c - 1} \Big\} \quad and \quad \tilde{\sigma}_{c,3}^2 = 2(1 - \frac{1}{c})r^4 + \frac{1}{p}\Big\{ \frac{2c^3\sigma^4}{(1-c)^4} + \frac{c\sigma^4(\nu_4 - 3)}{(c-1)^2} \Big\}.$$

*The corresponding* $(1 - \alpha)$*-confidence interval is given by*

$$P(L_{\alpha,c} \leq R_{\mathbf{X}, \boldsymbol{\beta}}(\hat{\boldsymbol{\beta}}, \boldsymbol{\beta}) \leq U_{\alpha,c}) \to 1 - \alpha, \tag{11}$$

*with*

$$L_{\alpha,c} = \frac{\sigma^2}{c_n - 1} + (1 - \frac{1}{c_n})r^2 + \frac{1}{\sqrt{p}}(\tilde{\mu}_{c,3} - Z_{\alpha/2}\tilde{\sigma}_{c,3}),$$

$$U_{\alpha,c} = \frac{\sigma^2}{c_n - 1} + (1 - \frac{1}{c_n})r^2 + \frac{1}{\sqrt{p}}(\tilde{\mu}_{c,3} + Z_{\alpha/2}\tilde{\sigma}_{c,3}).$$

**Remark 4.3.** *Note that besides the leading constants in* $(\mu_{c,3}, \sigma_{c,3})$*, the version* $(\tilde{\mu}_{c,3}, \tilde{\sigma}_{c,3})$ *also contains smaller order terms, including terms of order* $O(1/\sqrt{p})$ *in* $\tilde{\mu}_{c,3}$ *and terms of order* $O(1/p)$ *in* $\tilde{\sigma}_{c,3}$*. These smaller order terms will vanish when* $p$ *and* $n$ *grow very large, but for finite sample situations, these smaller order terms will provide a finer approximation for the finite sample distribution of* $R_{\mathbf{X}, \boldsymbol{\beta}}(\hat{\boldsymbol{\beta}}, \boldsymbol{\beta})$*. As shown in the following experiments, these terms have indeed made non-negligible contributions to fitting the empirical distribution of* $R_{\mathbf{X}, \boldsymbol{\beta}}(\hat{\boldsymbol{\beta}}, \boldsymbol{\beta})$*, which sheds new lights for practitioners.*

**Remark 4.4.** *If we compare the results in Theorem 4.3 and 4.5, we will find out that* $R_{\mathbf{X}}(\hat{\boldsymbol{\beta}}, \boldsymbol{\beta})$ *with constant* $\boldsymbol{\beta}$ *and* $R_{\mathbf{X}, \boldsymbol{\beta}}(\hat{\boldsymbol{\beta}}, \boldsymbol{\beta})$ *with random* $\boldsymbol{\beta}$ *share the same first-order limit and second-order error rate* $O(p^{-1/2})$*. This is quite intuitive because both risk treat* $\boldsymbol{\beta}$ *as a constant. Their differences are reflected in their limiting variances. Nevertheless, it's very interesting to observe from Theorem 4.4 that,* $R_{\mathbf{X}}(\hat{\boldsymbol{\beta}}, \boldsymbol{\beta})$ *with random* $\boldsymbol{\beta}$ *under the over-parametrized case has a smaller second-order error rate* $O(p^{-1})$*. It enjoys the same rate as the under-parametrized case in Theorem 4.1. A possible explanation would be that averaging over the randomness in* $\boldsymbol{\beta}$ *can partially offset the curse of dimensionality so that* $R_{\mathbf{X}}(\hat{\boldsymbol{\beta}}, \boldsymbol{\beta})$ *achieves the same error rate for all* $p, n$ *combinations.*

**Remark 4.5.** *It's worth mentioning that the only assumption regarding data distribution is assumption* (A)*, where only finite fourth order moment is required. Non-Gaussianity allows our theoretical results more widely applied.*

### 4.4 DISCUSSION

In this section, we first make a short conclusion of what we have done theoretically in this paper and further discuss some possible directions of extension.

We have systematically investigated the second-order fluctuations of two types of prediction risk, $R_{\mathbf{X}}(\hat{\boldsymbol{\beta}}, \boldsymbol{\beta})$ and $R_{\mathbf{X},\beta}(\hat{\boldsymbol{\beta}}, \boldsymbol{\beta})$, for the high dimensional least square estimators $\hat{\boldsymbol{\beta}}$. Theorem 4.1 and 4.4 are for $R_{\mathbf{X}}(\hat{\boldsymbol{\beta}}, \boldsymbol{\beta})$ while Theorem 4.2 and 4.5 are for $R_{\mathbf{X},\beta}(\hat{\boldsymbol{\beta}}, \boldsymbol{\beta})$. Both fixed effect and random effect of the regression coefficients $\boldsymbol{\beta}$ are discussed following the settings in Hastie et al. (2019). $R_{\mathbf{X}}(\hat{\boldsymbol{\beta}}, \boldsymbol{\beta})$ and $R_{\mathbf{X},\beta}(\hat{\boldsymbol{\beta}}, \boldsymbol{\beta})$ are the same when $\boldsymbol{\beta}$ is nonrandom as established in Theorem 4.3. Asymptotic results are categorized into the under-parametrized case ($p < n$) and the over-parametrized case ($p > n$).

Although the first-order limits of the prediction risk in high dimensional linear models have already been well studied in recent years, including general extensions to anisotropic features and signals in Wu & Xu (2020), the *"double descent"* risk curve is just a function of the limiting ratio $\lim p/n$. There is still a non-negligible discrepancy between the finite-sample prediction risk and its first-order limit on the *"double descent"* curve. How large is this discrepancy? How fast does the risk converge to its limit? Our CLTs provide answers to such questions and give a fine-grained characterization of the second-order fluctuations of the prediction risks. Not only explicit forms of the leading constants in the limiting means and variances are shown in the main theorems, smaller order terms are also derived to improve the empirical performance for practitioners.

It is also important to recognize the limitations of our results. First, the present paper only concerns linear regression tasks since the linear regression task is simple but important as well. Some recent works linearize neural networks at the initialization and employ Neural Tangent Kernels (Jacot et al. (2018)) to approximate the training procedure of a strongly over-parametrized neural network by solving a linear regression task, e.g. Du et al. (2018); Arora et al. (2019); Lee et al. (2019). Though the setting considered in this paper is simple and limited, the problem has not however been fully understood so far in the literature. Therefore, we are among the first to take the task and develop the second-order fluctuation results for the prediction risk. Second, we assume general covariance $\boldsymbol{\Sigma}$ and non-Gaussianity for the under-parametrized case, which fits the most updated and realistic settings in the literature, however we only investigate the isotropic settings while still allow for non-Gaussianity under over-parametrization. We haven't extended it to the more general anisotropic settings yet. The reasons are two-fold. On the one hand, according to Wu & Xu (2020), the first-order limits depend on the Stieltjes transforms of the unknown spectral distribution of $\boldsymbol{\Sigma}$. Since $\boldsymbol{\Sigma}$ is unknown, we cannot obtain any explicit characterization of the first-order limits, not to mention the second-order fluctuations. The CLTs would only be written as certain complicated implicit functions of $\boldsymbol{\Sigma}$ and would be too abstract to evaluate practically. More restrictions would be imposed on $\boldsymbol{\Sigma}$ to guarantee the second-order convergence. On the other hand, from the technical perspective, the techniques required for anisotropic over-parametrized cases are very different from the isotropic cases due to difference in the bias-variance decomposition in (4). The tools in random matrix theory have not been fully developed yet for anisotropic cases. Since we have considered various scenarios in this paper, including random and nonrandom signals $\boldsymbol{\beta}$ for both conditional and unconditional risks, it will take great efforts and continuous work to extend all of them to the most general settings, which would lead to many subsequent works in the field of machine learning and random matrix theory literature.

## 5 EXPERIMENTS

In this section, we carry out simulation experiments to examine the central limit theorems and the corresponding confidence intervals in Theorem 4.2 and Theorem 4.5. We generate data points from the linear model (1) and directly compute the prediction risk via the bias-variance decomposition in (4). The generative distribution $P_{\mathbf{x}}$ is taken to be the standard normal distribution. The noise distribution $P_\epsilon$ is taken to be $N(0, 1)$. In the following, we present the gap between the finite-sample distribution of the prediction risk and the corresponding limiting distribution to check the central limit theorems and use the cover rate to measure the effectiveness of the confidence intervals. More simulation results, including cases with non-Gaussian distributions for $P_{\mathbf{x}}$ and $P_\epsilon$ are relegated to the Appendix due to space limitations.

**Example 1**. This example examines the results in Theorem 4.2. We define a statistic

$$T_n = \frac{p}{\sigma_c}\left(R_{\mathbf{X},\boldsymbol{\beta}}(\hat{\boldsymbol{\beta}}, \boldsymbol{\beta}) - \sigma^2 \frac{c_n}{1 - c_n}\right) - \frac{\mu_c}{\sigma_c}.$$

According to Theorem 4.2, $T_n$ weakly converges to the standard normal distribution as $n, p \to \infty$. In this example, $c = 1/2$ and $p = 50, 100, 200$. The finite-sample distribution of $T_n$ is presented by the histogram of $T_n$ in Figure 2 with 1000 repetitions, where the solid blue curve stands for standard normal density function. It can be seen that the finite-sample distribution of $T_n$ is very consistent with the standard normal distribution, especially when $n, p$ become larger. When $\alpha = 0.05$, the empirical cover rates of the 95%-confidence interval are 94.2%, 93.5% and 95.3% for $p = 50, 100$ and 200, respectively. All these experiments verify the correctness of our theoretical results.



Figure 2: The histogram of $T_n$. The solid line is the density of the standard normal distribution.

**Example 2**. This example verifies the results in Theorem 4.5. Here we define two statistics:

$$T_{n,0} = \frac{\sqrt{p}}{\sigma_{c,3}}\left\{R_{\mathbf{X},\boldsymbol{\beta}}(\hat{\boldsymbol{\beta}}, \boldsymbol{\beta}) - (1 - \frac{1}{c_n})r^2 - \frac{\sigma^2}{c_n - 1}\right\} - \frac{\mu_{c,3}}{\sigma_{c,3}},$$

$$T_{n,1} = \frac{\sqrt{p}}{\tilde{\sigma}_{c,3}}\left\{R_{\mathbf{X},\boldsymbol{\beta}}(\hat{\boldsymbol{\beta}}, \boldsymbol{\beta}) - (1 - \frac{1}{c_n})r^2 - \frac{\sigma^2}{c_n - 1}\right\} - \frac{\tilde{\mu}_{c,3}}{\tilde{\sigma}_{c,3}}.$$

According to Theorem 4.5, both $T_{n,0}$ and $T_{n,1}$ weakly converge to the standard normal distribution as $n, p \to +\infty$. Compared to $T_{n,0}$, $T_{n,1}$ provides a better approximation for the finite sample distribution of $R_{\mathbf{X},\boldsymbol{\beta}}(\hat{\boldsymbol{\beta}}, \boldsymbol{\beta})$ because it contains smaller order terms in the asymptotic mean and variance. We take $c = 3/2$ and $p = 150, 300, 450$. Similarly the finite-sample distributions of $T_{n,0}$ and $T_{n,1}$ are presented by the histogram of $T_{n,0}$ and $T_{n,1}$ with 1000 repetitions. The comparison between these two statistics is shown in Figure 3. It can also be seen that the finite sample distributions of $T_{n,0}$ and $T_{n,1}$ both match the standard normal distribution quite well, especially $T_{n,1}$ with more precise characterization. The empirical cover rates of the 95%-confidence interval (11) are 93.8%, 94.7% and 94.4% for $p = 150, 300$ and 600 respectively, which further shows the validity of our theoretical results.

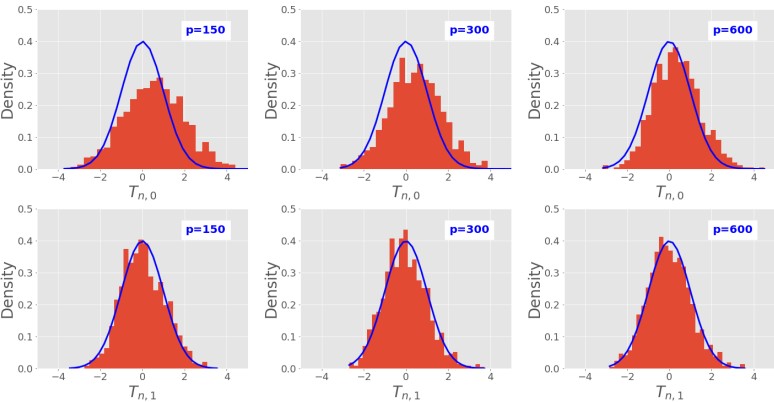

Figure 3: The histograms of $T_{n,0}$ and $T_{n,1}$. The solid line is the density of the standard normal distribution.

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

## A   PROOF OF THEOREM 4.1 AND THEOREM 4.2

Let $\mathbf{X} = \mathbf{Z}\boldsymbol{\Sigma}^{1/2}$. According to the Bai-Yin theorem (Bai & Yin (2008)), the smallest eigenvalue of $\mathbf{Z}^{\mathrm{T}}\mathbf{Z}/n$ is almost surely larger than $(1 - \sqrt{c})^2/2$ for sufficiently large $n$. Thus

$$\lambda_{min}(\frac{1}{n}\mathbf{X}^{\mathrm{T}}\mathbf{X}) \geq c_0\lambda_{\min}(\frac{1}{n}\mathbf{Z}^{\mathrm{T}}\mathbf{Z}) \geq \frac{c_0}{2}(1 - \sqrt{c})^2,$$

which implies that the matrix $\mathbf{X}^{\mathrm{T}}\mathbf{X}/n$ is almost surely invertible for large $n$. By Section 3.2, $\boldsymbol{\Pi} = \mathbf{0}$, $B_{\mathbf{X}}(\hat{\boldsymbol{\beta}}, \boldsymbol{\beta}) = B_{\mathbf{X},\boldsymbol{\beta}}(\hat{\boldsymbol{\beta}}, \boldsymbol{\beta}) = 0$ and $V_{\mathbf{X}}(\hat{\boldsymbol{\beta}}, \boldsymbol{\beta}) = V_{\mathbf{X},\boldsymbol{\beta}}(\hat{\boldsymbol{\beta}}, \boldsymbol{\beta})$. Thus the CLT of $R_{\mathbf{X}}(\hat{\boldsymbol{\beta}}, \boldsymbol{\beta})$ is same to that of $R_{\mathbf{X},\boldsymbol{\beta}}(\hat{\boldsymbol{\beta}}, \boldsymbol{\beta})$. For simplicity, we focus on $R_{\mathbf{X}}(\hat{\boldsymbol{\beta}}, \boldsymbol{\beta})$ in the following. Notice that

$$\begin{aligned}
V_{\mathbf{X}}(\hat{\boldsymbol{\beta}}, \boldsymbol{\beta}) &= \frac{\sigma^2}{n}\mathrm{Tr}(\hat{\boldsymbol{\Sigma}}^{-1}\boldsymbol{\Sigma}) \\
&= \frac{\sigma^2}{n}\mathrm{Tr}\left(\boldsymbol{\Sigma}^{-1/2}\big(\frac{\mathbf{Z}^{\mathrm{T}}\mathbf{Z}}{n}\big)^{-1}\boldsymbol{\Sigma}^{-1/2}\boldsymbol{\Sigma}\right) \\
&= \frac{\sigma^2}{n}\sum_{i=1}^{p}\frac{1}{s_i} = \frac{\sigma^2 p}{n}\int\frac{1}{s}dF_{\mathbf{Z}}(s)
\end{aligned}$$

where $F_{\mathbf{Z}}$ is the spectral measure of $\mathbf{Z}^{\mathrm{T}}\mathbf{Z}/n$. According to Theorem 1 of Hastie et al. (2019), as $n, p \to \infty$ such that $p/n = c_n \to c \in (0, \infty)$, $F_{\mathbf{Z}}(x)$ weakly converges to the standard Marcenko-Pastur law $F_c(x)$ and

$$V_{\mathbf{X}}(\hat{\boldsymbol{\beta}}, \boldsymbol{\beta}) \to \sigma^2 c\int\frac{1}{s}dF_c(s) = \sigma^2\frac{c}{1-c}.$$

Here the standard Marcenko-Pastur law $F_c(x)$ has a density function

$$p_c(x) = \begin{cases} \frac{1}{2\pi cx}\sqrt{(b-x)(x-a)}, & \text{if } a \leq x \leq b, \\ 0, & \text{o.w.}, \end{cases}$$

where $a = (1 - \sqrt{c})^2$, $b = (1 + \sqrt{c})^2$ and $p_c(x)$ has a point mass $1 - \frac{1}{c}$ at the origin if $c > 1$. Hence

$$\begin{aligned}
R_{\mathbf{X}}(\hat{\boldsymbol{\beta}}, \boldsymbol{\beta}) - \sigma^2\frac{c_n}{1-c_n} &= \frac{\sigma^2 p}{n}\int\frac{1}{s}dF_Z(s) - \sigma^2 c_n\int\frac{1}{s}dF_{c_n}(s) \\
&= \sigma^2 c_n\int\frac{1}{s}\big(dF_Z(s) - dF_{c_n}(s)\big).
\end{aligned}$$

According to Theorem 1.1 of Bai & Silverstein (2004),

$$p\left(R_{\mathbf{X}}(\hat{\boldsymbol{\beta}}, \boldsymbol{\beta}) - \sigma^2 \frac{c_n}{1 - c_n}\right) \xrightarrow{d} N(\mu_c, \sigma_c^2),\tag{12}$$

where

$$\mu_c = -\frac{\sigma^2 c}{2\pi i} \oint_\gamma \frac{1}{z} \frac{c\underline{m}(z)^3(1 + \underline{m}(z))^{-3}}{\{1 - c\underline{m}(z)^2(1 + \underline{m}(z))^{-2}\}^2} dz \tag{13}$$
$$-\frac{\sigma^2 c(\nu_4 - 3)}{2\pi i} \oint_\gamma \frac{1}{z} \frac{c\underline{m}(z)^3(1 + \underline{m}(z))^{-3}}{1 - c\underline{m}(z)^2(1 + \underline{m}(z))^{-2}} dz,$$

$$\sigma_c^2 = -\frac{\sigma^4 c^2}{2\pi^2} \oint_{\mathcal{C}_1} \oint_{\mathcal{C}_2} \frac{1}{z_1 z_2} \frac{1}{(\underline{m}(z_1) - \underline{m}(z_2))^2} \frac{d}{dz_1}\underline{m}(z_1)\frac{d}{dz_2}\underline{m}(z_2) dz_1 dz_2 \tag{14}$$
$$-\frac{\sigma^4 c^3(\nu_4 - 3)}{4\pi^2} \oint_{\mathcal{C}_1} \oint_{\mathcal{C}_2} \frac{1}{z_1 z_2} \frac{1}{(1 + \underline{m}(z_1))^2(1 + \underline{m}(z_2))^2} d\underline{m}(z_1) d\underline{m}(z_2).$$

Here the contours in (13) and (14) are closed and taken in the positive direction in the complex plane, enclosing the support of $F_{\mathbf{Z}}$, i.e. $[(1 - \sqrt{c})^2, (1 + \sqrt{c})^2]$. The Stieltjes transform $\underline{m}(z)$ satisfies the equation

$$z = -\frac{1}{\underline{m}} + \frac{c}{1 + \underline{m}}.$$

To further simplify the integrations in $\mu_c$ and $\sigma_c$, let $z = 1 + \sqrt{c}(r\xi + \frac{1}{r\xi}) + c$ and perform change of variables, then we have

$$\underline{m}(z) = -\frac{1}{1 + \sqrt{c}r\xi}, \quad dz = \sqrt{c}(r - \frac{1}{r\xi^2})d\xi, \quad d\underline{m} = \frac{\sqrt{c}r}{(1 + \sqrt{c}r\xi)^2}d\xi$$

and when $\xi$ moves along the unit circle $|\xi| = 1$ on the complex plane, $z$ will orbit around the center point $1 + c$ along an ellipse which enclosing the support of $F_{\mathbf{Z}}$. Thus

$$\mu_c = -\frac{\sigma^2 c}{2\pi i} \oint_\gamma \frac{1}{z} \frac{c\underline{m}(z)^3(1 + \underline{m}(z))^{-3}}{(1 - c\underline{m}(z)^2(1 + \underline{m}(z))^{-2})^2} dz$$
$$-\frac{\sigma^2 c(\nu_4 - 3)}{2\pi i} \oint_\gamma \frac{1}{z} \frac{c\underline{m}(z)^3(1 + \underline{m}(z))^{-3}}{1 - c\underline{m}(z)^2(1 + \underline{m}(z))^{-2}} dz$$
$$= \frac{\sigma^2 c}{2\pi i} \oint_{|\xi|=1} \frac{1}{r(\sqrt{c} + r\xi)(1 + \sqrt{c}r\xi)(\xi - \frac{1}{r})(\xi + \frac{1}{r})} d\xi$$
$$+\frac{\sigma^2 c(\nu_4 - 3)}{2\pi i} \oint_{|\xi|=1} \frac{1}{r\xi^2(\sqrt{c} + r\xi)(1 + \sqrt{c}r\xi)} d\xi$$
$$= \frac{\sigma^2 c^2}{(c - 1)^2} + \frac{\sigma^2 c^2(\nu_4 - 3)}{1 - c}.$$

As for $\sigma_c^2$, note that

$$\frac{1}{2\pi i} \oint_{\gamma_1} \frac{1}{z_1(\underline{m}_1 - \underline{m}_2)^2} d\underline{m}_1$$
$$= \frac{1}{2\pi i} \oint_{|\xi_1|=1} \frac{1}{1 + \sqrt{c}(r_1\xi_1 + \frac{1}{r_1\xi_1}) + c} \cdot \frac{\sqrt{c}\,r_1}{(\underline{m}_2 + \frac{1}{1+\sqrt{c}r_1\xi_1})^2(1 + \sqrt{c}r_1\xi_1)^2} d\xi_1$$
$$= \frac{1}{2\pi i} \oint_{|\xi_1|=1} \frac{\sqrt{c}\,r_1\xi_1}{(\xi_1 + \frac{\sqrt{c}}{r_1})(r_1\xi_1\sqrt{c} + 1)\{(r_1\xi_1\sqrt{c} + 1)\underline{m}_2 + 1\}^2} d\xi_1$$
$$= \frac{c}{(c - 1)\{(c - 1)\underline{m}_2 - 1\}^2},$$

therefore

$$
\begin{aligned}
&-\frac{\sigma^4 c^2}{2\pi^2} \oiint \frac{1}{z_1 z_2 (\underline{m}_1 - \underline{m}_2)^2} d\underline{m}_1 d\underline{m}_2 \\
&= \frac{2\sigma^4 c^2}{2\pi i} \oint_{|\xi_2|=1} \frac{c}{z_2(c-1)\left\{(c-1)\underline{m}_2 - 1\right\}^2} d\underline{m}_2 \\
&= \frac{2\sigma^4 c^2}{2\pi i} \oint_{|\xi_2|=1} \frac{\sqrt{c}\, r_2^2 \xi_2}{(c-1)(1+\sqrt{c}\, r_2 \xi_2)(\sqrt{c}+r_2\xi_2)^3} d\xi_2 = \frac{2c^3 \sigma^4}{(c-1)^4}.
\end{aligned}
$$

Meanwhile,

$$
\begin{aligned}
&\frac{1}{2\pi i} \oint_{\gamma_1} \frac{1}{z_1} \frac{1}{(1+\underline{m}(z_1))^2} d\underline{m}(z_1) \\
&= \frac{1}{2\pi i} \oint_{|\xi|=1} \frac{1}{\sqrt{c}\xi(1+\sqrt{c}r\xi)(\sqrt{c}+r\xi)} d\xi = \frac{1}{c-1},
\end{aligned}
$$

hence

$$
-\frac{\sigma^4 c^3 (\nu_4 - 3)}{4\pi^2} \oint_{\mathcal{C}_1} \oint_{\mathcal{C}_2} \frac{1}{z_1 z_2} \frac{1}{(1+\underline{m}(z_1))^2(1+\underline{m}(z_2))^2} d\underline{m}(z_1) d\underline{m}(z_2) = \frac{\sigma^4 c^3 (\nu_4 - 3)}{(1-c)^2},
$$

and

$$
\sigma_c^2 = \frac{2c^3 \sigma^4}{(c-1)^4} + \frac{\sigma^4 c^3 (\nu_4 - 3)}{(1-c)^2}.
$$

Let

$$
T_n = \frac{p}{\sigma_c}\left(R_{\mathbf{X}}(\hat{\boldsymbol{\beta}}, \boldsymbol{\beta}) - \sigma^2 \frac{c_n}{1-c_n} - \frac{\mu_c}{p}\right).
$$

According to (12), we have

$$
P(L_{\alpha,c} \leq R_{\mathbf{X},\boldsymbol{\beta}}(\hat{\boldsymbol{\beta}}, \boldsymbol{\beta}) \leq U_{\alpha,c}) = P(-Z_{\alpha/2} \leq T_n \leq Z_{\alpha/2}) \to 1 - \alpha,
$$

where

$$
\begin{aligned}
L_{\alpha,c} &= \sigma^2 \frac{c_n}{1-c_n} + \frac{1}{p}(\mu_c - Z_{\alpha/2}\sigma_c), \\
U_{\alpha,c} &= \sigma^2 \frac{c_n}{1-c_n} + \frac{1}{p}(\mu_c + Z_{\alpha/2}\sigma_c).
\end{aligned}
$$

$\square$

## B  PROOF OF THEOREM 4.3

Notice that

$$
\begin{aligned}
B_{\mathbf{X}}(\hat{\boldsymbol{\beta}}, \boldsymbol{\beta}) &= \boldsymbol{\beta}^{\mathrm{T}}(\boldsymbol{I}_p - \hat{\boldsymbol{\Sigma}}^+ \hat{\boldsymbol{\Sigma}})\boldsymbol{\beta} \\
&= \lim_{z\to 0^+} \boldsymbol{\beta}^{\mathrm{T}}\left(\boldsymbol{I}_p - (\hat{\boldsymbol{\Sigma}} + z\boldsymbol{I}_p)^{-1}\hat{\boldsymbol{\Sigma}}\right)\boldsymbol{\beta} \\
&= \lim_{z\to 0^+} z\boldsymbol{\beta}^{\mathrm{T}}(\hat{\boldsymbol{\Sigma}} + z\boldsymbol{I}_p)^{-1}\boldsymbol{\beta}.
\end{aligned}
$$

Since $\boldsymbol{\beta}$ is a constant vector, we can make use of the results in Theorem 3 in Bai et al. (2007) and Theorem 1.3 in Pan & Zhou (2008) regarding eigenvectors. Their works investigate the sample covariance matrix $\mathbf{A}_p = \boldsymbol{T}_p^{1/2} \mathbf{X}_p^{\mathrm{T}} \mathbf{X}_p \boldsymbol{T}_p^{1/2}/n$, where $\boldsymbol{T}_p$ is an $p \times p$ nonnegative definite Hermitian matrix with a square root $\boldsymbol{T}_p^{1/2}$ and $\mathbf{X}_p$ is an $n \times p$ matrix with i.i.d. entries $(\mathrm{x}_{ij})_{n \times p}$. Let $\boldsymbol{U}_p \boldsymbol{\Lambda}_p \boldsymbol{U}_p^{\mathrm{T}}$ denote the spectral decomposition of $\mathbf{A}_p$ where $\boldsymbol{\Lambda}_p = \mathrm{diag}(\lambda_1, \cdots, \lambda_p)$ and $\boldsymbol{U}_p$ is a unitary matrix consisting of the orthonormal eigenvectors of $\mathbf{A}_p$. Assume that $\boldsymbol{x}_p$ is an arbitrary nonrandom unit vector and $\boldsymbol{y} = (y_1, y_2, \cdots, y_p)^{\mathrm{T}} = \boldsymbol{U}_p^{\mathrm{T}} \boldsymbol{x}_p$, two empirical distribution functions based on eigenvectors and eigenvalues are defined as

$$
F_1^{\mathbf{A}_p}(x) = \sum_{i=1}^{p} |y_i|^2 \mathbb{1}(\lambda_i \leq x), \quad F^{\mathbf{A}_p}(x) = \frac{1}{p}\sum_{i=1}^{p} \mathbb{1}(\lambda_i \leq x).
$$

Then for a bounded continuous function $g(x)$, we have

$$\sum_{j=1}^{p} |\mathbf{y}_j|^2 g(\lambda_j) - \frac{1}{p}\sum_{j=1}^{p} g(\lambda_j) = \int g(x)dF_1^{\boldsymbol{A}_p}(x) - \int g(x)dF^{\boldsymbol{A}_p}(x).$$

The results in Bai et al. (2007) and Pan & Zhou (2008) show that

**Lemma B.1. (Theorem 3 Bai et al. (2007) and Theorem 1.3 Pan & Zhou (2008))** *Suppose that*

*(1)* $\mathbf{x}_{ij}$*'s are i.i.d. satisfying* $\mathbb{E}(\mathbf{x}_{ij}) = 0$, $\mathbb{E}(|\mathbf{x}_{ij}|^2) = 1$ *and* $\mathbb{E}(|\mathbf{x}_{ij}|^4) < \infty$;

*(2)* $\boldsymbol{x}_p \in \mathbb{C}^p$, $\|\boldsymbol{x}_p\| = 1$, $\lim_{n,p\to\infty} p/n = c \in (0,\infty)$;

*(3)* $\boldsymbol{T}_p$ *is nonrandom Hermitian non-negative definite with with its spectral norm bounded in* $p$, *with* $H_p = F^{\boldsymbol{T}_p} \xrightarrow{d} H$ *a proper distribution function and* $\boldsymbol{x}_p^{\mathrm{T}}(\boldsymbol{T}_p - z\boldsymbol{I}_p)^{-1}\boldsymbol{x}_p \to m_{F^H}(z)$, *where* $m_{FH}(z)$ *denotes the Stieltjes transform of* $H(t)$;

*(4)* $g_1, \cdots, g_k$ *are analytic functions on an open region of the complex plain which contains the real interval*

$$\left[\liminf_{p} \lambda_{min}(\boldsymbol{T}_p)\mathbb{1}_{(0,1)}(c)(1-\sqrt{c})^2, \ \limsup_{p} \lambda_{max}(\boldsymbol{T}_p)\mathbb{1}_{(0,1)}(c)(1+\sqrt{c})^2\right];$$

*(5) as* $n, p \to \infty$,

$$\sup_{z} \sqrt{n}\left\|\boldsymbol{x}_p^{\mathrm{T}}\big(\underline{m}_{F^{c_n}, H_p}(z)\boldsymbol{T}_p - \boldsymbol{I}_p\big)^{-1}\boldsymbol{x}_p - \int \frac{1}{1 + t\underline{m}_{F^{c_n}, H_p}(z)}dH_n(t)\right\| \to 0.$$

*Define* $G_p(x) = \sqrt{n}(F_1^{\boldsymbol{A}_p}(x) - F^{\boldsymbol{A}_p}(x))$, *then the random vectors*

$$\left(\int g_1(x)dG_p(x), \cdots, \int g_k(x)dG_p(x)\right)$$

*forms a tight sequence and converges weakly to a Gaussian vector* $\mathbf{x}_{g_1}, \cdots, \mathbf{x}_{g_k}$ *with mean zero and covariance function*

$$\mathrm{Cov}(\mathbf{x}_{g_1}, \mathbf{x}_{g_2}) = -\frac{1}{2\pi^2}\int_{\mathbb{C}_1}\int_{\mathbb{C}_2} g_1(z_1)g_2(z_2)\frac{(z_2\underline{m}_2 - z_1\underline{m}_1)^2}{c^2 z_1 z_2(z_2 - z_1)(\underline{m}_2 - \underline{m}_1)}dz_1 dz_2.$$

*The contours* $\mathbb{C}_1$, $\mathbb{C}_2$ *are disjoint, both contained in the analytic region for the functions* $(g_1, \cdots, g_k)$ *and enclose the support of* $F^{c_n, H_p}$ *for all large* $p$.

*(6) If* $H(x)$ *satisfies*

$$\int \frac{dH(t)}{(1 + t\underline{m}(z_1))(1 + t\underline{m}(z_2))} = \int \frac{dH(t)}{1 + t\underline{m}(z_1)}\int \frac{dH(t)}{1 + t\underline{m}(z_2)},$$

*then the covariance function can be further simplified to*

$$\mathrm{Cov}(\mathbf{x}_{g_1}, \mathbf{x}_{g_2}) = \frac{2}{c}\Big(\int g_1(x)g_2(x)dF^{c,H}(x) - \int g_1(x)dF^{c,H}(x)\int g_2(x)dF^{c,H}(x)\Big).$$

Recall that $B_{\mathbf{X}}(\hat{\boldsymbol{\beta}}, \boldsymbol{\beta}) = \lim_{z\to 0^+} z\boldsymbol{\beta}^{\mathrm{T}}(\hat{\boldsymbol{\Sigma}} + z\boldsymbol{I}_p)^{-1}\boldsymbol{\beta}$. Let $g(x) = 1/(x+z)$ and $\boldsymbol{x}_p = \boldsymbol{\beta}/r$. Then we have

$$\int g(x)dG_n(x) = \sqrt{n}\Big(\frac{1}{r^2}\boldsymbol{\beta}^{\mathrm{T}}(\hat{\boldsymbol{\Sigma}} + z\boldsymbol{I}_p)^{-1}\boldsymbol{\beta} - \int g(x)dF_{c_n}(x)\Big),$$

where $F_{c_n}(x)$ is the standard Marcenko-Pastur law. It is not difficult to check that under Assumptions (A1), (B1) and (C1), all the conditions *(1)-(6)* in Lemma B.1 are satisfied.

To proceed further, denote $a = (1-\sqrt{c})^2$, $b = (1+\sqrt{c})^2$. If $c$ is replaced by $c_n$, $a$ and $b$ are denoted by $a_n$ and $b_n$ respectively. By some algebraic calculations, we have

$$\begin{aligned}
\int g(x)dF_{c_n}(x) &= (1-\frac{1}{c_n})\cdot\frac{1}{z} + \int_{a_n}^{b_n}\frac{1}{x+z}\cdot\frac{1}{2\pi c_n x}\sqrt{(b_n - x)(x - a_n)}dx \\
&= (1-\frac{1}{c_n})\cdot\frac{1}{z} - \frac{-1 + c_n + z - \sqrt{c_n^2 + 2c_n(z-1) + (1+z)^2}}{2c_n z},
\end{aligned}$$

and

$$\mathrm{Var}(\mathrm{x}_g) = \frac{2}{c}\left(\int\{g(x)\}^2 dF_c(x) - \left\{\int g(x)dF_c(x)\right\}^2\right)$$

$$= \frac{2}{c}\left((1-\frac{1}{c})\cdot\frac{1}{z^2} + \int_a^b \frac{1}{(x+z)^2}\cdot\frac{1}{2\pi cx}\sqrt{(b-x)(x-a)}dx\right)$$

$$-\frac{2}{c}\left((1-\frac{1}{c})\cdot\frac{1}{z} + \int_a^b \frac{1}{x+z}\cdot\frac{1}{2\pi cx}\sqrt{(b-x)(x-a)}dx\right)^2.$$

Therefore,

$$\lim_{z\to0^+} z\int g(x)dF_{c_n}(x) = 1 - \frac{1}{c_n} \quad\text{and}\quad \lim_{z\to0^+} z^2\mathrm{Var}(\mathrm{x}_g) = \frac{2(c-1)}{c^3}.$$

Furthermore, as $n, p \to \infty$, $p/n = c_n \to c > 1$,

$$\sqrt{n}\left(B_{\mathbf{X}}(\hat{\boldsymbol{\beta}},\boldsymbol{\beta}) - (1-\frac{1}{c_n})r^2\right) \xrightarrow{d} N\left(0, \frac{2(c-1)}{c^3}r^4\right).$$

This can be rewritten as

$$\sqrt{p}\left(B_{\mathbf{X}}(\hat{\boldsymbol{\beta}},\boldsymbol{\beta}) - (1-\frac{1}{c_n})r^2\right) \xrightarrow{d} N\left(0, \frac{2(c-1)}{c^2}r^4\right).$$

Next we deal with the variance term $V_{\mathbf{X}}(\hat{\boldsymbol{\beta}},\boldsymbol{\beta})$. According to the Assumption *(B1)*, the variance term is

$$V_{\mathbf{X}}(\hat{\boldsymbol{\beta}},\boldsymbol{\beta}) = \frac{\sigma^2}{n}\mathrm{Tr}\{\hat{\boldsymbol{\Sigma}}^+\} = \frac{\sigma^2}{n}\sum_{i=1}^n \frac{1}{s_i},$$

where $s_i$, $i = 1,\ldots,n$ are the nonzero eigenvalues of $\mathbf{X}^\mathrm{T}\mathbf{X}/n$. Let $\{t_i,\ i = 1,\ldots n\}$ denote the non-zero eigenvalues of $\mathbf{X}\mathbf{X}^\mathrm{T}/p$, then we have

$$V_{\mathbf{X}}(\hat{\boldsymbol{\beta}},\boldsymbol{\beta}) = \frac{\sigma^2}{p}\sum_{i=1}^n \frac{1}{t_i} = \frac{\sigma^2 n}{p}\int\frac{1}{t}dF_{\mathbf{X}\mathbf{X}^\mathrm{T}/p}(t) \to \frac{\sigma^2}{c-1}.$$

By interchanging the role of $p$ and $n$, from the result in Theorem 4.1, as $n, p \to \infty$, $p/n = c_n \to c > 1$, we have,

$$\sum_{i=1}^n \frac{1}{t_i} - \frac{n}{1-c_n'} \xrightarrow{d} N\left(\frac{c'}{(c'-1)^2} + \frac{c'(\nu_4-3)}{1-c'}, \frac{2c'}{(c'-1)^4} + \frac{c'(\nu_4-3)}{(1-c')^2}\right).$$

where $c_n' = n/p = 1/c_n$, $c' = 1/c$. This result can be rewritten as

$$\sum_{i=1}^n \frac{1}{t_i} - \frac{p}{c_n-1} \xrightarrow{d} N\left(\frac{c}{(1-c)^2} + \frac{(\nu_4-3)}{c-1}, \frac{2c^3}{(1-c)^4} + \frac{c(\nu_4-3)}{(c-1)^2}\right).$$

Hence the CLT of $V_{\mathbf{X}}(\hat{\boldsymbol{\beta}},\boldsymbol{\beta})$ is given by

$$p\left(V_{\mathbf{X}}(\hat{\boldsymbol{\beta}},\boldsymbol{\beta}) - \frac{\sigma^2}{c_n-1}\right) \xrightarrow{d} N\left(\frac{c\sigma^2}{(1-c)^2} + \frac{\sigma^2(\nu_4-3)}{c-1}, \frac{2c^3\sigma^4}{(1-c)^4} + \frac{c\sigma^4(\nu_4-3)}{(c-1)^2}\right).$$

Notice that $\mathrm{Cov}\left(B_{\mathbf{X}}(\hat{\boldsymbol{\beta}},\boldsymbol{\beta}),\ V_{\mathbf{X}}(\hat{\boldsymbol{\beta}},\boldsymbol{\beta})\right) = 0$. According to the consistency rate and the limiting distribution of $B_{\mathbf{X}}(\hat{\boldsymbol{\beta}},\boldsymbol{\beta})$ and $V_{\mathbf{X}}(\hat{\boldsymbol{\beta}},\boldsymbol{\beta})$, we know that the bias $B_{\mathbf{X}}(\hat{\boldsymbol{\beta}},\boldsymbol{\beta})$ is the leading term of $R_{\mathbf{X}}(\hat{\boldsymbol{\beta}},\boldsymbol{\beta})$. This implies that

$$\sqrt{p}\left\{R_{\mathbf{X}}(\hat{\boldsymbol{\beta}},\boldsymbol{\beta}) - (1-\frac{1}{c_n})\|\boldsymbol{\beta}\|_2^2 - \frac{\sigma^2}{c_n-1}\right\} \xrightarrow{d} N\left(0, \sigma_{c,1}^2\right),$$

where $\sigma_{c,1}^2 = 2(c-1)r^4/c^2$. A practical version of this CLT is given by

$$\sqrt{p}\left\{R_{\mathbf{X}}(\hat{\boldsymbol{\beta}},\boldsymbol{\beta}) - (1-\frac{1}{c_n})\|\boldsymbol{\beta}\|_2^2 - \frac{\sigma^2}{c_n-1}\right\} \xrightarrow{d} N\left(\tilde{\mu}_{c,1}, \tilde{\sigma}_{c,1}^2\right),$$

where

$$\tilde{\mu}_{c,1} = \frac{1}{\sqrt{p}}\left\{\frac{c\sigma^2}{(1-c)^2} + \frac{\sigma^2(\nu_4-3)}{c-1}\right\},$$

$$\tilde{\sigma}_{c,1}^2 = \frac{2(c-1)}{c^2}r^4 + \frac{1}{p}\left\{\frac{2c^3\sigma^4}{(1-c)^4} + \frac{c\sigma^4(\nu_4-3)}{(c-1)^2}\right\}.$$

## C    PROOF OF THEOREM 4.4

First we consider the bias term $B_{\mathbf{X}}(\hat{\boldsymbol{\beta}}, \boldsymbol{\beta})$. By Assumption (A1), (B1), and (C2),

$$
\begin{aligned}
B_{\mathbf{X}}(\hat{\boldsymbol{\beta}}, \boldsymbol{\beta}) &= \mathbb{E}[\boldsymbol{\beta}^{\mathrm{T}}\Pi\boldsymbol{\Sigma}\Pi\boldsymbol{\beta}|\mathbf{X}] = \mathbb{E}[\boldsymbol{\beta}^{\mathrm{T}}\Pi\boldsymbol{\beta}|\mathbf{X}] \\
&= \mathrm{Tr}\left\{(\boldsymbol{I}_p - \hat{\boldsymbol{\Sigma}}^{+}\hat{\boldsymbol{\Sigma}})\mathbb{E}(\boldsymbol{\beta}\boldsymbol{\beta}^{\mathrm{T}}|\mathbf{X})\right\} \\
&= \frac{r^2}{p}\mathrm{Tr}\{\boldsymbol{I}_p - \hat{\boldsymbol{\Sigma}}^{+}\hat{\boldsymbol{\Sigma}}\} = r^2(1 - n/p).
\end{aligned}
$$

Alternatively, we can rewrite the bias as

$$
\begin{aligned}
B_{\mathbf{X}}(\hat{\boldsymbol{\beta}}, \boldsymbol{\beta}) &= \lim_{z \to 0^+} \mathbb{E}[\boldsymbol{\beta}^{\mathrm{T}}(\boldsymbol{I}_p - (\hat{\boldsymbol{\Sigma}} + z\boldsymbol{I}_p)^{-1}\hat{\boldsymbol{\Sigma}})\boldsymbol{\beta}|\mathbf{X}] \\
&= \lim_{z \to 0^+} \mathbb{E}[z\boldsymbol{\beta}^{\mathrm{T}}(\hat{\boldsymbol{\Sigma}} + z\boldsymbol{I}_p)^{-1}\boldsymbol{\beta}|\mathbf{X}] \\
&= \lim_{z \to 0^+} z\frac{r^2}{p}\mathrm{Tr}(\hat{\boldsymbol{\Sigma}} + z\boldsymbol{I}_p)^{-1}.
\end{aligned}
$$

Define that $f_n(z) = z\frac{r^2}{p}\mathrm{Tr}(\hat{\boldsymbol{\Sigma}} + z\boldsymbol{I}_p)^{-1}$. Notice that $|f_n(z)|$ and $|f'_n(z)|$ are bounded above. By the Arzela-Ascoli theorem, we deduce that $f_n(z)$ converges uniformly to its limit. Under Assumption *(C2)*, by the Moore-Osgood theorem, almost surely,

$$
\begin{aligned}
\lim_{n,p \to \infty} B_{\mathbf{X}}(\hat{\boldsymbol{\beta}}, \boldsymbol{\beta}) &= \lim_{z \to 0^+} \lim_{n,p \to \infty} z\frac{r^2}{p}\mathrm{Tr}(\hat{\boldsymbol{\Sigma}} + z\boldsymbol{I}_p)^{-1} \\
&= \lim_{z \to 0^+} \lim_{n,p \to \infty} z\frac{r^2}{p}\mathrm{Tr}\left(\frac{1}{n}\mathbf{X}\mathbf{X}^{\mathrm{T}} + z\boldsymbol{I}_n\right)^{-1},
\end{aligned}
$$

In fact,

$$
\lim_{n,p \to \infty} B_{\mathbf{X}}(\hat{\boldsymbol{\beta}}, \boldsymbol{\beta}) = r^2 \lim_{z \to 0^+} \lim_{n,p \to \infty} zm_n(-z),
$$

where $m_n(z)$ is the Stieltjes transform of empirical spectral distribution of $\hat{\boldsymbol{\Sigma}} = \mathbf{X}^{\mathrm{T}}\mathbf{X}/n$. According to Theorem 2.1 in Zheng et al. (2015) and Lemma 1.1 in Bai & Silverstein (2004), the truncated version of $p(m_n(z) - m(z))$ converges weakly to a two-dimensional Gaussian process $M(\cdot)$ satisfying

$$
\mathbb{E}[M(z)] = \frac{c\underline{m}^3(1 + \underline{m})}{\{(1 + \underline{m})^2 - c\underline{m}^2\}^2} + \frac{c(\nu_4 - 3)\underline{m}^3}{(1 + \underline{m})\{(1 + \underline{m})^2 - c\underline{m}^2\}},
$$

and

$$
\begin{aligned}
\mathrm{Cov}\big(M(z_1), M(z_2)\big) &= 2\left\{\frac{\underline{m}'(z_1)\underline{m}'(z_2)}{(\underline{m}(z_1) - \underline{m}(z_2))^2} - \frac{1}{(z_1 - z_2)^2}\right\} \\
&\quad + \frac{c(\nu_4 - 3)\underline{m}'(z_1)\underline{m}'(z_2)}{(1 + \underline{m}(z_1))^2(1 + \underline{m}(z_2))^2},
\end{aligned}
$$

where $\underline{m} = \underline{m}(z)$ represents the Stieltjes transform of limiting spectral distribution of companion matrix $\mathbf{X}\mathbf{X}^{\mathrm{T}}/n$ satisfying the equation

$$
z = -\frac{1}{\underline{m}} + \frac{c}{1 + \underline{m}}, \quad \underline{m}(z) = -\frac{1 - c}{z} + cm(z).
$$

When $p > n$, we can actually solve $\underline{m}(z)$ equation and obtain that

$$
\begin{aligned}
\underline{m}(z) &= \frac{-1 + c - z + \sqrt{-4z + (1 - c + z)^2}}{2z}, \\
m(z) &= \frac{1 - c - z + \sqrt{-4z + (1 - c + z)^2}}{2cz}.
\end{aligned}
$$

Therefore, by some algebraic calculations, we have

$$
\begin{aligned}
\lim_{n,p\to\infty} B_{\mathbf{X}}(\hat{\boldsymbol{\beta}}, \boldsymbol{\beta}) &= \lim_{n,p\to\infty} r^2 \lim_{z\to 0^+} z m_n(-z) = r^2 \lim_{z\to 0^+} \left\{ z m(-z) + z(1 - \frac{1}{c}) \frac{1}{z} \right\} \\
&= \lim_{n,p\to\infty} r^2 \lim_{z\to 0^+} z \frac{n}{p} \underline{m}_n(z) = r^2 \frac{1}{c} \lim_{z\to 0^+} z \underline{m}(-z) \\
&= r^2 (1 - \frac{1}{c}).
\end{aligned}
$$

Moreover,

$$
\begin{aligned}
\mathrm{Var}\big(M(z)\big) &= \lim_{z_1\to z_2 = z} \mathrm{Cov}\big(M(z_1), M(z_2)\big) \\
&= \frac{2\underline{m}'(z)\underline{m}'''(z) - 3(\underline{m}''(z))^2}{6(\underline{m}'(z))^2} + \frac{c(\nu_4 - 3)(\underline{m}'(z))^2}{(1 + \underline{m}(z))^4}.
\end{aligned}
$$

By substituting of the explicit form of $\underline{m}(z)$, we can easily derive that

$$
\lim_{z\to 0^+} z\mathbb{E}[M(-z)] = 0, \quad \lim_{z\to 0^+} z^2 \mathrm{Var}(M(-z)) = 0,
$$

which means that the second-order limit of $B_{\mathbf{X}}(\hat{\boldsymbol{\beta}}, \boldsymbol{\beta})$ is still $r^2(1 - 1/c)$. All in all, $B_{\mathbf{X}}(\hat{\boldsymbol{\beta}}, \boldsymbol{\beta})$ is identical with a constant $r^2(1 - 1/c)$ in distribution.

On the other hand, by Assumption (B1),

$$
V_{\mathbf{X}}(\hat{\boldsymbol{\beta}}, \boldsymbol{\beta}) = \frac{\sigma^2}{n} \mathrm{Tr}\{\hat{\boldsymbol{\Sigma}}^+\} = \frac{\sigma^2}{n} \sum_{i=1}^{n} \frac{1}{s_i},
$$

where $s_i$, $i = 1, \ldots, n$ are the nonzero eigenvalues of $\mathbf{X}^{\mathrm{T}}\mathbf{X}/n$. Similar to the proof of Theorem 4.3, the CLT of $V_{\mathbf{X}}(\hat{\boldsymbol{\beta}}, \boldsymbol{\beta})$ is given by

$$
p\Big(V_{\mathbf{X}}(\hat{\boldsymbol{\beta}}, \boldsymbol{\beta}) - \frac{\sigma^2}{c_n - 1}\Big) \xrightarrow{d} N\Big(\frac{c\sigma^2}{(1-c)^2} + \frac{\sigma^2(\nu_4 - 3)}{c - 1}, \frac{2c^3\sigma^4}{(1-c)^4} + \frac{c\sigma^4(\nu_4 - 3)}{(c-1)^2}\Big).
$$

Combining the results of $B_{\mathbf{X}}(\hat{\boldsymbol{\beta}}, \boldsymbol{\beta})$ and $V_{\mathbf{X}}(\hat{\boldsymbol{\beta}}, \boldsymbol{\beta})$, we have

$$
p\Big\{R_{\mathbf{X}}(\hat{\boldsymbol{\beta}}, \boldsymbol{\beta}) - r^2(1 - \frac{1}{c_n}) - \frac{\sigma^2}{c_n - 1}\Big\} \xrightarrow{d} N(\mu_{c,2}, \sigma_{c,2}^2),
$$

where

$$
\mu_{c,2} = \frac{c\sigma^2}{(1-c)^2} + \frac{\sigma^2(\nu_4 - 3)}{c - 1}, \quad \sigma_{c,2}^2 = \frac{2c^3\sigma^4}{(1-c)^4} + \frac{c\sigma^4(\nu_4 - 3)}{(c-1)^2}.
$$

## D   PROOF OF THEOREM 4.5

Note that under Assumption *(B1)* and *(C2)*, $B_{\mathbf{X}, \boldsymbol{\beta}}(\hat{\boldsymbol{\beta}}, \boldsymbol{\beta}) = \boldsymbol{\beta}^{\mathrm{T}}\boldsymbol{\Pi}\boldsymbol{\beta} = \boldsymbol{\beta}^{\mathrm{T}}(\boldsymbol{I}_p - \hat{\boldsymbol{\Sigma}}^+\hat{\boldsymbol{\Sigma}})\boldsymbol{\beta}$. If we directly consider $\boldsymbol{\beta}^{\mathrm{T}}(\boldsymbol{I}_p - \hat{\boldsymbol{\Sigma}}^+\hat{\boldsymbol{\Sigma}})\boldsymbol{\beta}$, we can make use of the asymptotic results for quadratic forms Theorem 7.2 in Bai & Yao (2008) stated as follows.

**Lemma D.1. (Theorem 7.2 in Bai & Yao (2008))** *Let $\{\boldsymbol{A}_n = [a_{ij}(n)]\}$ be a sequence of $n \times n$ real symmetric matrices, $\{\mathbf{x}_i\}_{i\in\mathbb{N}}$ be a sequence of i.i.d. $K$ dimensional real random vectors, with $\mathbb{E}(\mathbf{x}_i) = 0$, $\mathbb{E}(\mathbf{x}_i\mathbf{x}_i^{\mathrm{T}}) = (\gamma_{ij})_{K\times K}$ and $\mathbb{E}[\|\mathbf{x}_i\|^4] < \infty$. Denote*

$$
\mathbf{x}_i = (\mathrm{x}_{\ell i})_{K\times 1}, \quad \mathbf{X}(\ell) = (\mathrm{x}_{\ell 1}, \cdots, \mathrm{x}_{\ell n})^{\mathrm{T}}, \quad \ell = 1, \cdots, K, \ i = 1, \cdots, n,
$$

*assume the following limits exist*

$$
\omega = \lim_{n\to\infty} \frac{1}{n} \sum_{i=1}^{n} a_{ii}^2(n), \quad \theta = \lim_{n\to\infty} \frac{1}{n} \mathrm{Tr}\, \boldsymbol{A}_n^2.
$$

*Then the $K$-dimensional random vectors*

$$
\mathbf{z}_n = (\mathrm{z}_{n,\ell})_{K\times 1}, \quad \mathrm{z}_{n,\ell} = \frac{1}{\sqrt{n}}\big(\mathbf{X}(\ell)^{\mathrm{T}}\boldsymbol{A}_n\mathbf{X}(\ell) - \gamma_{\ell\ell}\mathrm{Tr}\{\boldsymbol{A}_n\}\big), \quad 1 \le \ell \le K,
$$

*converge weakly to a zero-mean Gaussian vector with covariance matrix $\boldsymbol{D} = \boldsymbol{D}_1 + \boldsymbol{D}_2$ where*

$$
[\boldsymbol{D}_1]_{\ell\ell'} = \omega\big\{\mathbb{E}(x_{\ell 1}^2 x_{\ell' 1}^2) - \gamma_{\ell\ell}\gamma_{\ell'\ell'}\big\}, \quad [\boldsymbol{D}_2]_{\ell\ell'} = (\theta - \omega)(\gamma_{\ell\ell'}\gamma_{\ell'\ell} + \gamma_{\ell\ell'}^2), \ 1 \le \ell, \ell' \le K.
$$

According to the results in Lemma D.1, let $\boldsymbol{A}_n = \Pi = \boldsymbol{I}_p - \hat{\boldsymbol{\Sigma}}^+\hat{\boldsymbol{\Sigma}}$, then we have, as $p \to \infty$,

$$\sqrt{p}\Big\{\boldsymbol{\beta}^{\mathrm{T}}\Pi\boldsymbol{\beta} - \frac{r^2}{p}\operatorname{Tr}(\Pi)\Big\} \xrightarrow{d} N(0, d^2 = d_1^2 + d_2^2),$$

where

$$\omega = \lim_{p\to\infty}\frac{1}{p}\sum_{i=1}^{p}\Pi_{ii}^2, \quad \theta = \lim_{p\to\infty}\frac{1}{p}\operatorname{Tr}(\Pi^2) = 1 - \frac{1}{c},$$

and

$$
\begin{aligned}
d_1^2 &= \omega\big\{\mathbb{E}(x_{\ell 1}^2 x_{\ell 1}^2) - \gamma_{\ell\ell}^2\big\} = \omega\big(\frac{p^2}{r^4}\mathbb{E}(\beta_i^4) - 1\big)r^4, \\
d_2^2 &= (\theta - \omega)(\gamma_{\ell\ell}^2 + \gamma_{\ell\ell}^2) = 2(\theta - \omega)r^4.
\end{aligned}
$$

Since in the proof of Theorem 4.4, we have already shown that

$$\frac{r^2}{p}\operatorname{Tr}(\Pi) = r^2(1 - \frac{n}{p}).$$

In particular, if $\boldsymbol{\beta}$ follows multivariate Gaussian distribution, i.e. $\boldsymbol{\beta} \sim N_p(0, \frac{r^2}{p}\boldsymbol{I}_p)$, then as $p \to \infty$,

$$\sqrt{p}\Big\{B_{\mathbf{X},\boldsymbol{\beta}}(\hat{\boldsymbol{\beta}}, \boldsymbol{\beta}) - r^2(1 - \frac{n}{p})\Big\} \xrightarrow{d} N\Big(0, 2(1 - \frac{1}{c})r^4\Big).$$

Moreover, $V_{\mathbf{X},\boldsymbol{\beta}}(\hat{\boldsymbol{\beta}}, \boldsymbol{\beta}) = V_{\mathbf{X}}(\hat{\boldsymbol{\beta}}, \boldsymbol{\beta})$, we have already proved in Theorem 4.4 that

$$p(V_{\mathbf{X},\boldsymbol{\beta}}(\hat{\boldsymbol{\beta}}, \boldsymbol{\beta}) - \frac{\sigma^2}{c_n - 1}) \xrightarrow{d} N\Big(\frac{c\sigma^2}{(1-c)^2} + \frac{\sigma^2(\nu_4 - 3)}{c - 1}, \frac{2c^3\sigma^4}{(1-c)^4} + \frac{c\sigma^4(\nu_4 - 3)}{(c - 1)^2}\Big).$$

Note that $\operatorname{Cov}(B_{\mathbf{X},\boldsymbol{\beta}}(\hat{\boldsymbol{\beta}}, \boldsymbol{\beta}), V_{\mathbf{X},\boldsymbol{\beta}}(\hat{\boldsymbol{\beta}}, \boldsymbol{\beta})) = 0$. According to the consistency rate of $B_{\mathbf{X},\boldsymbol{\beta}}(\hat{\boldsymbol{\beta}}, \boldsymbol{\beta})$ and $V_{\mathbf{X},\boldsymbol{\beta}}(\hat{\boldsymbol{\beta}}, \boldsymbol{\beta})$, we know that the bias $B_{\mathbf{X}}(\hat{\boldsymbol{\beta}}, \boldsymbol{\beta})$ is the leading term of $R_{\mathbf{X},\boldsymbol{\beta}}(\hat{\boldsymbol{\beta}}, \boldsymbol{\beta})$. This implies that

$$\sqrt{p}\Big\{R_{\mathbf{X},\boldsymbol{\beta}}(\hat{\boldsymbol{\beta}}, \boldsymbol{\beta}) - r^2(1 - \frac{1}{c_n}) - \frac{\sigma^2}{c_n - 1}\Big\} \xrightarrow{d} N(0, \sigma_{c,3}^2),$$

where $\sigma_{c,3}^2 = 2r^4(1 - 1/c)$. A practical version of this CLT is given by

$$\sqrt{p}\Big\{R_{\mathbf{X},\boldsymbol{\beta}}(\hat{\boldsymbol{\beta}}, \boldsymbol{\beta}) - r^2(1 - \frac{1}{c_n}) - \frac{\sigma^2}{c_n - 1}\Big\} \xrightarrow{d} N(\tilde{\mu}_{c,3}, \tilde{\sigma}_{c,3}^2),$$

where

$$
\begin{aligned}
\tilde{\mu}_{c,3} &= \frac{1}{\sqrt{p}}\Big\{\frac{c\sigma^2}{(1-c)^2} + \frac{\sigma^2(\nu_4 - 3)}{c - 1}\Big\}, \\
\tilde{\sigma}_{c,3}^2 &= 2(1 - \frac{1}{c})r^4 + \frac{1}{p}\Big\{\frac{2c^3\sigma^4}{(1-c)^4} + \frac{c\sigma^4(\nu_4 - 3)}{(c - 1)^2}\Big\}.
\end{aligned}
$$

# E  MORE EXPERIMENTS

## E.1  MORE RESULTS OF EXAMPLE 1

This example checks Theorem 4.2. We define a statistic

$$T_n = \frac{p}{\sigma_c}\Big(R_{\mathbf{X}}(\hat{\boldsymbol{\beta}}, \boldsymbol{\beta}) - \sigma^2\frac{c_n}{1 - c_n}\Big) - \frac{\mu_c}{\sigma_c}.$$

According to Theorem 4.2, $T_n$ weakly converges to the standard normal distribution as $n, p \to \infty$. In this example, $c = 1/2$ and $p = 50, 100, 200$. To make sure the assumption (A) holds, the generative distribution $P_{\mathbf{x}}$ is taken to be the standard normal distribution, the centered gamma with shape $4.0$ and scale $0.5$, and the normalized Student-t distribution with $6.0$ degree of freedom. The finite-sample distribution of $T_n$ is estimated by the histogram of $T_n$ under 1000 repetitions. The results are presented in Figure 4. One can find that the finite-sample distribution of $T_n$ tends to the standard normal distribution as $n, p \to +\infty$. When $\alpha = 0.05$, the empirical cover rates of the 95%-confidence interval are reported in Figure 5.

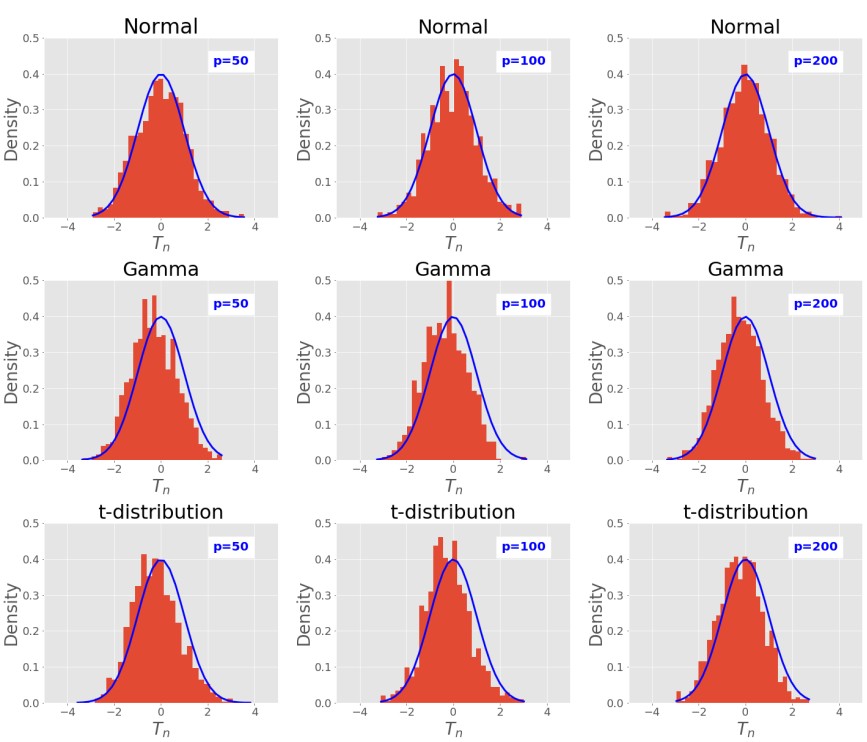

Figure 4: The histogram of $T_n$. The solid line is the density of the standard normal distribution.

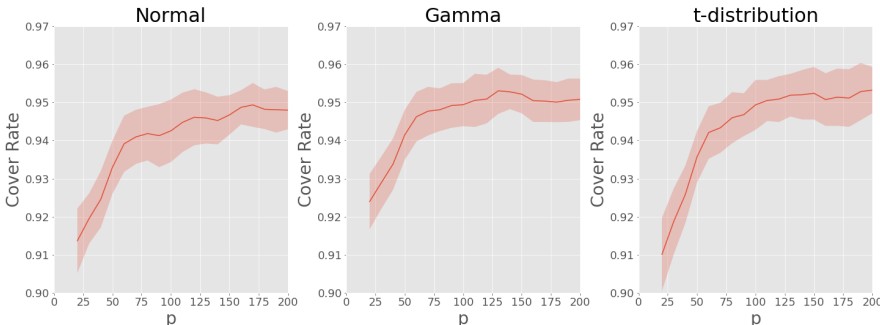

Figure 5: The cover rate of the confidence interval (7) as $p$ creases. The confidence level is $95\%$.

## E.2 MORE RESULTS OF EXAMPLE 2

The Example 2 checks Theorem 4.5. Here we consider the standardized statistics:

$$
\begin{aligned}
T_{n,0} &= \frac{\sqrt{p}}{\sigma_{c,3}}\left\{R_{\mathbf{X}}(\hat{\boldsymbol{\beta}},\boldsymbol{\beta}) - (1-\frac{1}{c_n})r^2 - \frac{\sigma^2}{c_n-1}\right\} - \frac{\mu_{c,3}}{\sigma_{c,3}}, \\
T_{n,1} &= \frac{\sqrt{p}}{\tilde{\sigma}_{c,3}}\left\{R_{\mathbf{X}}(\hat{\boldsymbol{\beta}},\boldsymbol{\beta}) - (1-\frac{1}{c_n})r^2 - \frac{\sigma^2}{c_n-1}\right\} - \frac{\tilde{\mu}_{c,3}}{\tilde{\sigma}_{c,3}}.
\end{aligned}
$$

According to the central limit theorem (10) and its practical version, both $T_{n,0}$ and $T_{n,1}$ weakly converge to the standard normal distribution as $n,p \to +\infty$. We take $c = 2$ and $p = 100, 200, 400$. The finite-sample distributions of $T_{n,0}$ and $T_{n,1}$ are estimated by the histogram of $T_{n,0}$ and $T_{n,1}$ under 1000 repetitions. The results are presented in Figure 6 and Figure 7. When $\alpha = 0.05$, the empirical cover rates of the 95%-confidence interval (11) are reported in Figure 8.

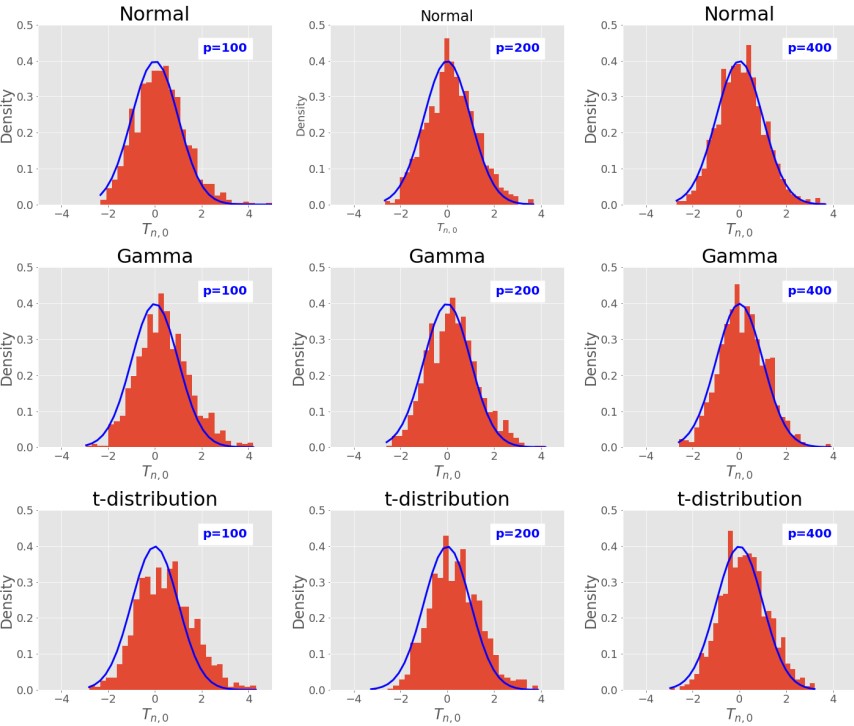

Figure 6: The histogram of $T_{n,0}$. The solid line is the density of the standard normal distribution.

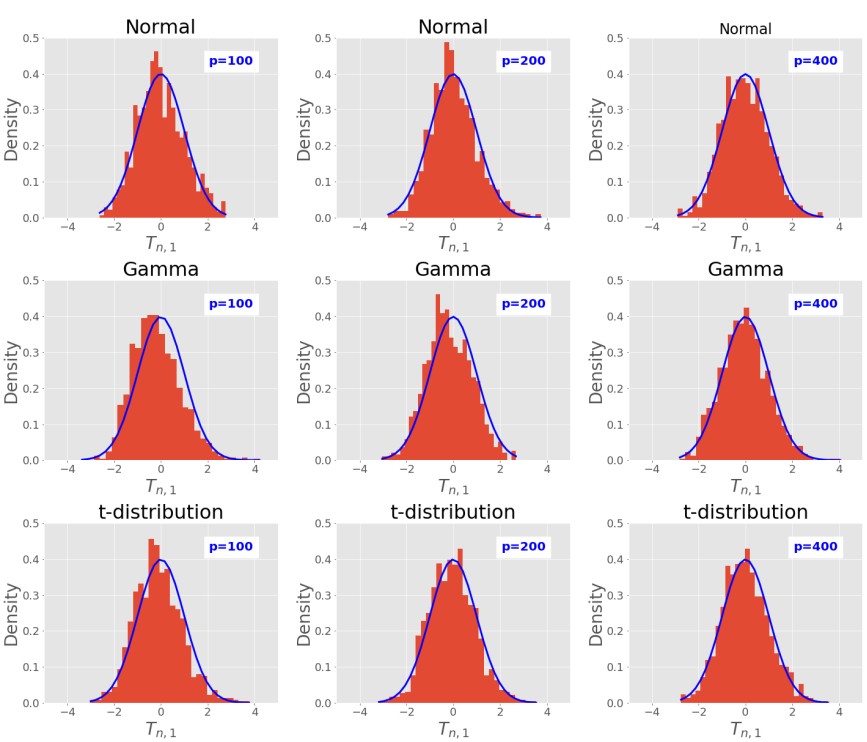

Figure 7: The histogram of $T_{n,1}$. The solid line is the density of the standard normal distribution.

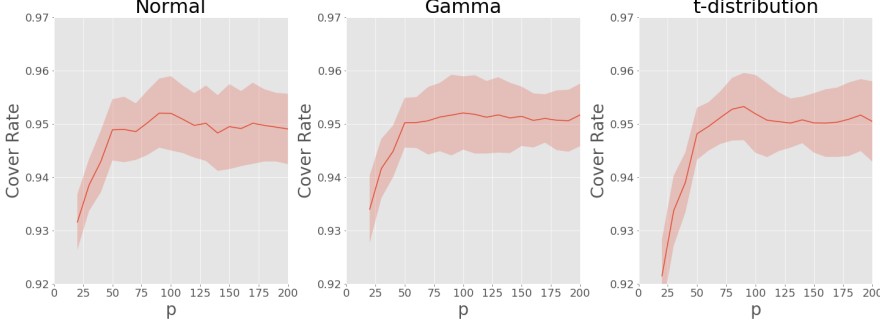

Figure 8: The cover rate of the confidence interval (11) as $p$ creases. The confidence level is $95\%$.

## E.3 EXAMPLE 3

This example checks Theorem 4.3. To proceed further, we denote two statistics:

$$
\begin{aligned}
T_{n,2} &= \frac{\sqrt{p}}{\sigma_{c,1}}\left\{ R_{\mathbf{X}}(\hat{\boldsymbol{\beta}}, \boldsymbol{\beta}) - (1 - \frac{1}{c_n})r^2 - \frac{\sigma^2}{c_n - 1}\right\} - \frac{\mu_{c,1}}{\sigma_{c,1}}, \\
T_{n,3} &= \frac{\sqrt{p}}{\tilde{\sigma}_{c,1}}\left\{ R_{\mathbf{X}}(\hat{\boldsymbol{\beta}}, \boldsymbol{\beta}) - (1 - \frac{1}{c_n})r^2 - \frac{\sigma^2}{c_n - 1}\right\} - \frac{\tilde{\mu}_{c,1}}{\tilde{\sigma}_{c,1}}.
\end{aligned}
$$

According to the central limit theorem (8) and its practical version, both $T_{n,2}$ and $T_{n,3}$ weakly converge to the standard normal distribution as $n, p \to +\infty$. We take $c = 2$ and $p = 100, 200, 400$. The finite-sample distributions of $T_{n,2}$ and $T_{n,3}$ are estimated by the histogram of $T_{n,2}$ and $T_{n,3}$ under 1000 repetitions. The results are presented at Figure 9 and Figure 10. One can see that the finite-sample distributions of $T_{n,2}$ and $T_{n,3}$ are close to the standard normal distribution, and the finite-sample performance of $T_{n,3}$ is better than that of $T_{n,2}$. When $\alpha = 0.05$, the empirical cover rates of the 95%-confidence interval (9) are reported in Figure 11.

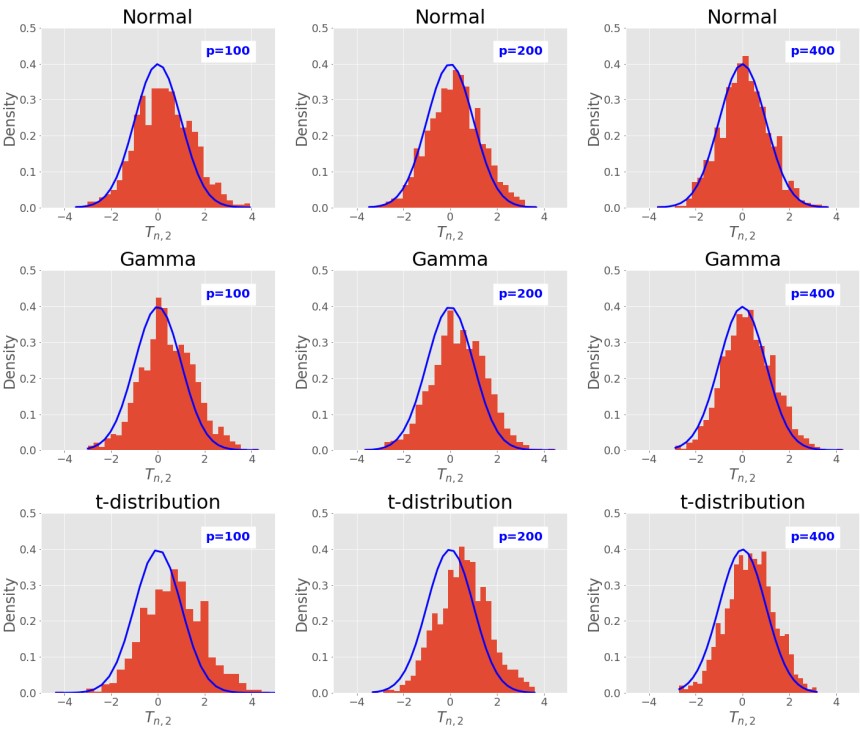

Figure 9: The histogram of $T_{n,2}$. The solid line is the density of the standard normal distribution.

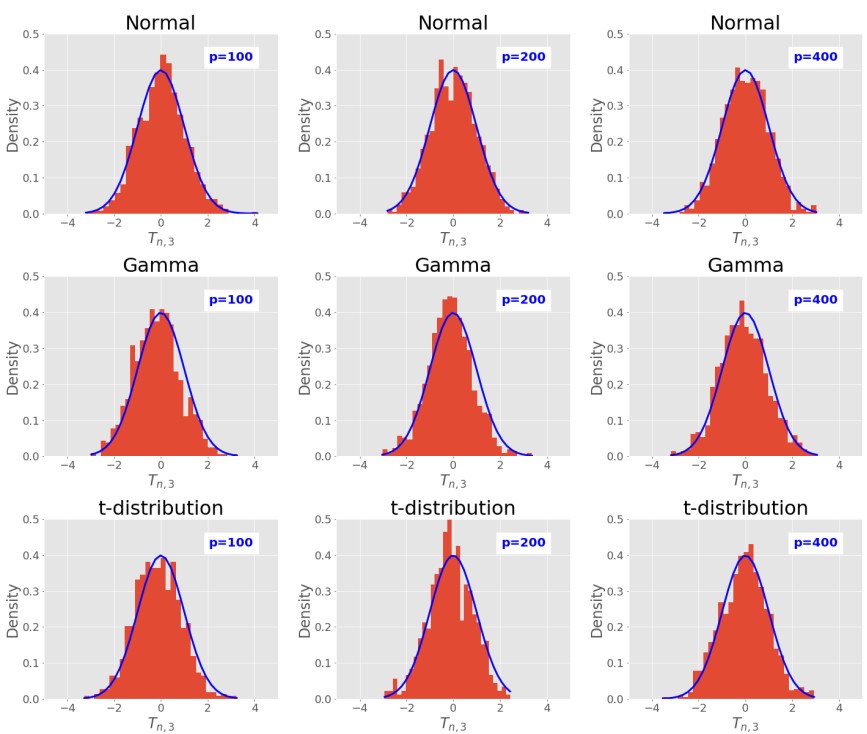

Figure 10: The histogram of $T_{n,3}$. The solid line is the density of the standard normal distribution.

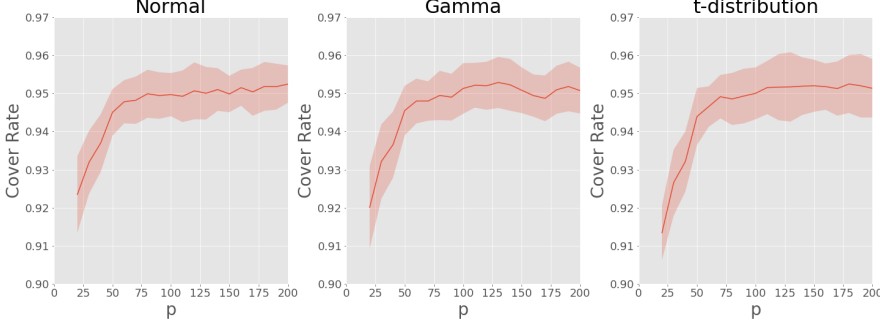

Figure 11: The coverage of confidence interval (9) as $p$ increases. The confidence level is $95\%$.

## F    AN EXAMPLE FROM FIGURE 1

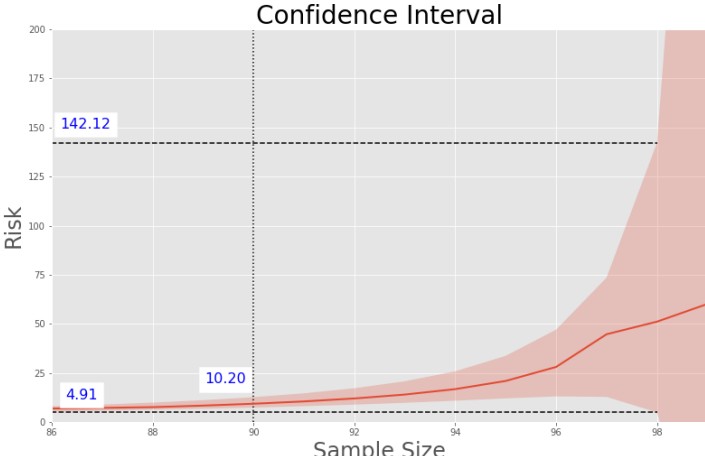

Figure 12: According to the first-order limit, given a fixed dimension $p = 100$, the prediction risks at sample size $n = 90$ and $n = 98$ are about $10.20$ and $49.02$. More data hurt seems true. However, the $95\%$ confidence interval of the prediction risks with sample size 98 is $[4.91, 142.12]$, which contains the risk for $n = 90$. Then more data hurt is not statistically significant.

## G    AN ANISOTROPIC EXAMPLE FOR REMARK 4.2

In the over-parameterized case, the bias term $B_{\mathbf{X}}(\hat{\boldsymbol{\beta}}, \boldsymbol{\beta}) = \boldsymbol{\beta}^{\mathrm{T}} \Pi \Sigma \Pi \boldsymbol{\beta}$ is non-zero while the variance term $V_{\mathbf{X}}(\hat{\boldsymbol{\beta}}, \boldsymbol{\beta})$ remains the same as under-parameterized case. Therefore in this section, we conduct a small simulation to examine the fluctuation of the bias $B_{\mathbf{X}}$ for both isotropic and anisotropic $\Sigma$ in the over-parameterized case with non-random $\boldsymbol{\beta}$ satisfying Assumption (C1). In particular, in the following we set $r = 1$.

We consider both localized and delocalized $\boldsymbol{\beta}$ such that

1. Localized case: $\boldsymbol{\beta}_1 = (1, 0, \cdots, 0)$;
2. Delocalized case: $\boldsymbol{\beta}_2 = \frac{1}{\sqrt{p}}(1, \cdots, 1)$;

and both the isotropic and anisotropic $\Sigma$

3. Identity case: $\Sigma_1 = \boldsymbol{I}_p$;
4. Compound symmetric case: $\Sigma_2 = 0.5\boldsymbol{I}_p + 0.5\boldsymbol{1}_p \boldsymbol{1}_p^{\mathrm{T}}$.

Then we fix $p/n = 2$ and let $p$ vary from 10 to 300, we present in Figure 13 the empirical variance of $\sqrt{p} * B_{\mathbf{X}}$ and $p * B_{\mathbf{X}}$ under various combinations of $\Sigma$ and $\boldsymbol{\beta}$ with 1000 replications.

From the plot on the top left panel in Figure 13, we can see that the variance of $\sqrt{p} * B_{\mathbf{X}}$ for both $\boldsymbol{\beta}_1$ and $\boldsymbol{\beta}_2$ remain constant as $p$ grows, which indicates that the convergence rate of $B_{\mathbf{X}}$ is $1/\sqrt{p}$ under the isotropic case regardless of localized or delocalized $\boldsymbol{\beta}$. As for the anisotropic case on the top right corner, the variance of $\sqrt{p} * B_{\mathbf{X}}$ stabilizes for $\boldsymbol{\beta}_1$, while decays for $\boldsymbol{\beta}_2$, which indicates that convergence rate of $B_{\mathbf{X}}$ under $(\Sigma_2, \boldsymbol{\beta}_2)$ and $(\Sigma_2, \boldsymbol{\beta}_1)$ are different.

This simulation result further confirms our conjecture that in the over-parameterized case, there is no universal CLT for the prediction risk $R_{\mathbf{X}}(\hat{\boldsymbol{\beta}}, \boldsymbol{\beta})$ under the anisotropic setting for non-random $\boldsymbol{\beta}$.

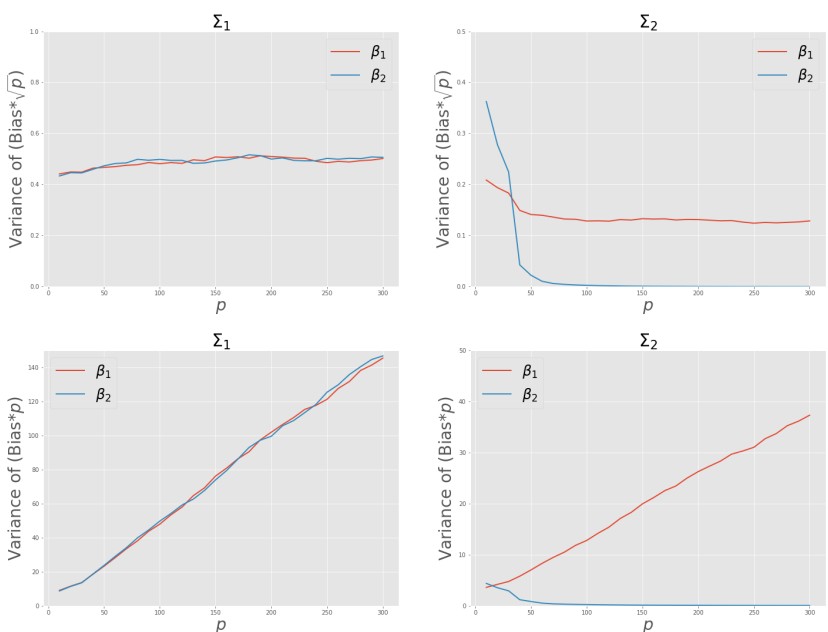

Figure 13: The upper panel is the empirical variance of $\sqrt{p} * B_{\mathbf{X}}$, the lower panel is for $p * B_{\mathbf{X}}$.

