# OpenReview forum: "Provable More Data Hurt in High Dimensional Least Squares Estimator"
_ICLR.cc/2021/Conference — Reject_

### Official Review · AnonReviewer4 · 2020-10-19
**Solid contribution in characterizing the second-order fluctuation of high-dimensional least squares estimators**

**Rating:** 7
**Confidence:** 4

**Review:**

**Summary**: In this article, the authors characterized the second-order fluctuation of the prediction risk of the (min-norm) least square estimator, by assuming an underlying noisy teacher model $y_i = \beta^T x_i + \epsilon_i$, in the regime where the data dimension $p$ and the number of training samples $n$ grow large at the same pace. Results in both the under-parameterized (Theorem 4.1 and 4.2) and over-parameterized regimes (Theorem 4.3-4.5) were provided, under the statistical model where the data $x_i$ are zero-mean random vectors with **generic** i.i.d. entries and then "rotated" to have some possible covariance structures. Numerical experiments for relatively small values of $n,p$ were conducted to support the theoretical assessment.

**Strong points**: The authors provided a solid contribution to the theoretical understanding of high-dimensional (over-parameterized) machine learning systems that are of growing interest today. The technical tool of the CLT for linear spectral statistics introduced here is popular in RMT literature and may be of independent interest to the machine learning community. To the best of my knowledge, there are very few results on the second-order fluctuations of the prediction, one possibly relevant paper is "Asymptotic normality and confidence intervals for derivatives of 2-layers neural network in the random features model" at NeurIPS 2020.

**Weak points**: This article could be strengthened by summarizing more explicitly the lessons to be learned for practitioners.

**Recommendation**: This is a good paper that made solid contributions to the theoretical understanding of high-dimensional least squares estimators and consequently, shed interesting light on the future design of more elaborate machine learning systems. I thus recommend it for publication at ICLR.

**Detailed comments**:

* Sec 1 introduction: "However, the existing asymptotic results, which focus on the first order limit of the prediction risk, cannot exactly guarantee the more data hurt phenomenon": it would be helpful to provide a more concrete illustrating example for the insufficiency of the first-order analysis, or perhaps simply refer to the discussion above Figure 1 below.
* The x- and y-axes are hardly visible in Figure 1, and the same applies to other figures in this article as well.
* below (1): is the independence between the entries of $\mathbf{x}_i$ necessary, should this be stated here?
* Theorem 4.3, 4.4 and 4.5: it would be helpful to comment here that only the case of **identity data covariance** is considered here, is the general covariance case easily follows, with just more complicated expressions, or there may be some technical challenge to master?
* Since the theoretical results in the paper are "universal" with respect to the distribution of the entries of $\mathbf{x}$, it would be good to provide numerical experiments on non-normal data to better illustrate the contribution of this work, or at least, to mention explicitly that experiments **on more general data distribution** are available in the Appendix.
* In Figure 3, we observe that the statistic $T_{n,0}$ fits less well the theoretical prediction (at least compared to $T_n$ in Figure 2 or $T_{n,1}$ in Figure 3), could the authors provide any theoretical justification or intuition on this?


**After rebuttal**:  I've read the authors' feedback and my score remains the same.

---

> ### Author Response · Authors · 2020-11-17
> **Response**
>
> Thank you so much for the comments and suggestions. The paper has been carefully revised according to your comments, and all changes are marked in red.  The point-by-point responses to the comments are presented as follows.
>
> Q1. "*To the best of my knowledge, there are very few results on ... ... , one possibly relevant paper is ... ...*"
>
> **Ans:** Thank you for this comment.  Indeed Shen & Bellec (2020) is a relevant paper which we have missed in the original version. We have cited it and added some related comments in Section 2 now.
>
> Q2. "*Sec 1 introduction ... ... it would be helpful to provide a more concrete illustrating example ... ...*"
>
> **Ans:** Thanks for the suggestion. We have added on some comments highlighted in red in the Introduction and a Figure 12 in Appendix F for illustration.
>
> Q3. "*The x- and y-axes are hardly visible in Figure 1 ... ...* "
>
> **Ans:** Thank you for the suggestion. We have enlarged the axis fonts in all figures in the new version.
>
> Q4. "*below (1): is the independence between the entries of* $x_i$ *necessary ... ...*"
>
> **Ans:** Thanks for the question. The independence assumption across $i=1, \ldots,n$ indeed indicates that our data are independently collected. However, we do not necessarily require the entries of each $x_i$ to be independent. The covariance between the entries of $x_i$ is Sigma. We didn’t impose any assumptions on Sigma here because Sigma is assumed to follow different structures, i.e. Assumption (B1) and (B2), in the main theorems.
>
> Q5. "*Theorem 4.3, 4.4 and 4.5: it would be helpful to comment here that ... ...*"
>
> **Ans:** Thanks for the comment. Indeed, in the under-parametered case, we allow the covariance matrix of $x_i$ to be general while in the over-parametered case, we only consider the identity covariance matrix. The reason that we haven’t extended it to the more general anisotropic settings is twofold. First, the first-order limits for anisotropic cases depend on the Stieltjes transforms of the unknown spectral distribution of $\Sigma$, see Wu & Xu (2020). Since $\Sigma$ is unknown, we cannot obtain any explicit characterization of the first-order limits, not to mention the second-order fluctuations. The CLTs would only be written as certain complicated implicit functions of $\Sigma$ and would be too abstract to evaluate practically. Second, from the technical perspective, the techniques required for anisotropic settings in the overparametrized cases are quite different from the isotropic cases due to the difference in the bias-variance decomposition. The tools in random matrix theory have not been fully developed yet to cover such anisotropic cases. In fact, since we have considered various scenarios in this paper, including random and nonrandom signals $\beta$ for two types of conditional prediction risk, it will take great efforts and requires continuous work to extend all of them to the most general settings, which would lead to many subsequent works in the field of machine learning and random matrix theory literature. We have added a new conclusion section 4.4 in this revision. Please refer to Section 4.4 for more discussions.
>
> Q6. "*... ... it would be good to provide numerical experiments on non-normal data to better illustrate the contribution of this work, or at least, ... ...*"
>
> **Ans:** Thanks for pointing it out. Our theoretical results do cover non-Gaussian cases and numerical results on non-Gaussian data are provided in the Appendix. In the revision, we have mentioned it in Section 5 for readers’ easy reference.
>
> Q7. "*In Figure 3, we observe that ...... could the authors provide any theoretical justification or intuition on this?*"
>
> **Ans:** Thanks for the question. Compared to $T_{n,0}$, $T_{n,1}$ provides a better approximation for the finite sample distribution of $R_{X,\beta}$. Because aside from the leading constants in the asymptotic mean and variance in $T_{n,0}$, $T_{n,1}$ also contains smaller order terms, including terms of order $O(1/\sqrt{p})$ in the mean and terms of order $O(1/p)$ in the variance. These smaller order terms will vanish when $p$ and $n$ become very large. However, for relatively small sample sizes, they can help build up a finer description of the distribution of $R_{X,\beta}$. We have added one remark below Theorem 4.5 and some comments in Example 2 for a brief explanation.
>
> Q8: “*This article could be strengthened by summarizing more explicitly the lessons to be learned for practitioners.*”
>
> **Ans:** Thanks for the comment. We have added a new discussion Section 4.4 and a new Remark 4.2 in this revision for this point.
>
> **Reference**
>
> Shen, Y. & Bellec, P. C. (2020). Asymptotic normality and confidence intervals for derivatives of 2-layers neural network in the random features model. Advances in Neural Information Processing Systems, 33.
>
> Wu, D. & Xu, J. (2020). On the optimal weighted l2 regularization in overparametrized linear regression. arXiv preprint arXiv:2006.05800, 2020.

---

### Official Review · AnonReviewer2 · 2020-10-27
**A good addition to double descent literature. Weak Accept because of the setting of isotropic features or signals.**

**Rating:** 6
**Confidence:** 4

**Review:**

This paper provides the central limit theorem type of results for generalization error of high-dimensional least squares estimator.

Strength: It is nice to know that the finite sample prediction error converges to its asymptotic limits in the CLT style. Previous results have shown that the prediction error converges to the asymptotic limit, but it is unclear how fast the convergence speed. This CLT type of results provides the answer and this explains why empirical simulations have shown good accuracy for a sample size of just 300-500. I think it is a nice piece of result to be added to the current double descent literature.

Weakness: One major concern is although CLT type of results is interesting, whether the overall level of contribution of the paper meets the standard of acceptance or in other words whether the story of CLT is complete. This paper seems to be on the borderline and the story looks to be only half-written. First, this paper only focused on the cases when features are isotropic or signals are isotropic. Recent double descent works have extended this setting to both anisotropic features and anisotropic signals (e.g. https://arxiv.org/abs/2006.05800) and those settings are more realistic and more interesting. Secondly, although $\sqrt{n}$ convergence is probably the optimal speed, it is good to have a lower bound result to rigorously show it. The story of this paper will be more complete if the authors can also show results on these two points.

In summary, I like the CLT type of result, but I recommend a weak accept because of the simple setting. I will give an 8 if the authors can extend their results to the most updated settings in this linear regression model and complete the lower bound results. Further, I think the authors can simplify their theorems and assumption quite a lot once they study the general anisotropic settings. It gives better clarity.

---

> ### Author Response · Authors · 2020-11-17
> **Response**
>
> Thank you for the insightful comments and suggestions. The paper has been carefully revised according to your comments, and all changes are marked in red.  The point-by-point responses to your comments are presented as follows.
>
> Q1. "*First, this paper only focused on the cases when features are isotropic or signals are isotropic. Recent double descent works have extended this setting to both anisotropic features and anisotropic signals (e.g. https://arxiv.org/abs/2006.05800) and those settings are more realistic and more interesting.*"
>
> **Ans:** Thank you for this suggestion.  The main asymptotic results in our paper have been categorized into the under-parametrized case  ($p < n$) in Section 4.2 and the over-parametrized case ($p > n$) in Section 4.3. For the under-parametrized case, both anisotropic and non-Gaussian features are allowed in the high dimensional linear models.
>
> As for the overparametrized case, indeed we have only studied the cases when either the features or the signals are isotropic. We haven’t extended it to the more general anisotropic settings yet in this paper. The reasons are two-fold. On the one hand, according to Theorem 3 of Hastie et al. (2019) and Wu & Xu (2020), even the first-order limit of the prediction risk does not always have explicit analytic forms under the anisotropic features. They depend on the Stieltjes transforms of the unknown spectral distribution of $\Sigma$. Since $\Sigma$ is unknown, we cannot obtain any explicit characterization of the first-order limits, not to mention second-order fluctuations. The CLTs would only be written as certain complicated implicit functions of $\Sigma$ and would be too abstract to evaluate pratically.
>
> On the other hand, from the technical perspective, the techniques required for anisotropic settings in the overparametrized cases are quite different from the isotropic settings due to different bias-variance decompositions. While the tools in random matrix theory have not been fully developed yet to cover the anisotropic cases. More restrictions would be imposed on $\Sigma$ to guarantee the second-order convergence. In fact, since we have considered various scenarios in this paper, including random and nonrandom signals $\beta$ for two types of conditional prediction risk, it will take great efforts and continuous work to extend all of the results to the most general settings, which would lead to many subsequent works in the field of both machine learning and random matrix theory literature.
>
> In this revised version, we have cited related works on anisotropic settings and discuss more on this point. Please see Section 4.4 for more discussions.
>
> Q2. "*Secondly, although* $\sqrt{n}$ *convergence is probably the optimal speed, it is good to have a lower bound result to rigorously show it. The story of this paper will be more complete if the authors can also show results on these two points.*"
>
> **Ans:** Thanks for the suggestion. In fact, the upper and lower bounds of the prediction risks have already been derived in the literature, please see Theorem 1 of Bartlett et al. (2020) for the lower bound result. The Central Limit Theorems derived in this paper have already answered the questions raised, which provide more sharp asymptotic results and more precise characterization of the second-order fluctuations. Based on these CLTs, we can not only learn how fast the prediction risk converges to its first order limit but also learn the distribution the discrepancies follow. Based on such CLT results, we can further derive confidence intervals where the finite sample prediction risk lie in, which fulfils the functionality of the lower and upper bounds.
>
>
> **Reference**
>
> Bartlett, P. L., Long, P. M., Lugosi, G., & Tsigler, A. (2020). Benign overfitting in linear regression. Proceedings of the National Academy of Sciences.
>
> Hastie, T., Montanari, A., Rosset, S., & Tibshirani, R. J. (2019). Surprises in high-dimensional ridgeless least squares interpolation. arXiv preprint arXiv:1903.08560.
>
> Wu, D., & Xu, Ji. (2020). On the optimal weighted l_2 regularization in overparametrized linear regression. arXiv preprint arXiv:2006.06800v4

---

> > ### Comment · AnonReviewer2 · 2020-11-21
> > **Discussions**
> >
> > About the isotropic setting, I agree that the Stieltjes transform does not have a specific form but rather a solution to a fixed equation when feature is aisotropic. But it is not an issue in practice that \Sigma is not known because neither the norm of true coefficients nor the noise variance is known as well. In the end, I think it is ok to have an implicit form of CLT. The important thing is to confirm the convergence speed is still 1/sqrt{p}. In other words, it is even ok to just confirm that the difference between the risk and its asymptotic limit is at the order of at most 1/\sqrt{p}.
> >
> > About convergence speed, in the newest version of Bartlett's paper, I do not find the Theorem 1. Also, I do not think Bartlett's paper has addressed the lower bound of the convergence speed as the prediction risk converging to its asymptotic limit. Their results are lower bound and upper bound on the risk itself. Specifically, for a converging sequence  A_n = C+o(1), I would like to have a lower bound on the o(1) term rather than the lower bound on C.

---

> > > ### Author Response · Authors · 2020-11-24
> > > **Reply**
> > >
> > > Thank you for continuing the discussion!
> > >
> > > 1. *"About the isotropic setting, ...... The important thing is to confirm the convergence speed is still* $1/\sqrt{p}$. *......"*
> > >
> > > **Ans:** Thank you for your insightful comment.  We agree that it is very important to confirm the convergence rate for anisotropic settings when $p>n$. We carefully looked into this problem and realized that the situations for nonrandom and random signal $\beta$ are quite different.
> > >
> > > As for the case with random $\beta$ considered in Theorem 4.4, the convergence rate for anisotropic case is the same with isotropic case, i.e. $1/p$. It’s because that the prediction risk $R_X$ can be decomposed into the bias term $B_X=(r^2/p)\text{tr}(\Pi \Sigma)$ (See Lemma 4 in Hastie et.al(2019)) and the variance term $V_X=(\sigma^2/n)\text{tr}(\hat{\Sigma}^+ \Sigma)$. According to random matrix theory, we can foresee that both $B_X$ and $V_X$ have convergence rate $1/p$, so as $R_X$.  In fact, Wu & Xu (2020) also considered the random $\beta$ case.
> > >
> > > As for the nonrandom $\beta$ case considered in Theorem 4.3 and 4.5, since the variance component $V_X$ remains the same with Theorem 4.4, the dominating convergence rate $1/\sqrt{p}$ originates from the bias term $B_X=\beta^T\Pi\Sigma\Pi\beta$. Under the isotropic settings when $\Sigma=I_p$, the eigenvector of $\hat\Sigma$ is asymptotically Haar distributed. Thanks to the good properties of Haar distribution, $B_X$ is only related to the length of $\beta$, i.e. $\beta^T\beta=r^2$. However, in the anisotropic settings with general $\Sigma$, the eigenvector of the sample covariance matrix is no longer asymptotically Haar distributed. The limiting behavior of $B_X$ heavily relies on the interaction between $\beta$ and the eigenvectors of the sample covariance matrix corresponding to $\Sigma$. Therefore, we think that there is no universal convergence rate for arbitrary $\beta$ and $\Sigma$. Some additional assumptions are required (both on $\beta$ and $\Sigma$) to guarantee the existence for its CLT. We have added a new Remark 4.2 to clarify this point and a small simulation is added in Appendix G for empirical illustration.
> > >
> > > In particular,  we compare the empirical variance of $B_X$ when $\beta$ is localized and delocalized respectively, i.e. $\beta=(1,0,…,0)$  and $\beta=(1/\sqrt{p})(1,1,..,1)$.  It can be seen from the simulation results that the two variances are not in the same scale at all, which indicates that the convergence rate under these two settings are different. Indeed the difference between the prediction risk and its asymptotic limit for two cases we considered both have smaller order than $O(1/\sqrt{p})$. We still need more theoretical justification for your insightful conjecture. However we can confirm that there is no universal convergence rate for the bias term that can cover arbitrary non-random $\beta$ and anisotropic $\Sigma$ in the over-parametrized case, not to mention the prediction risk. We will investigate this problem in our future work.
> > >
> > > As for the “implicit” issue, we are sorry that our previous explanations for the practical purpose were not clear enough. One of our main objectives is to prove the "More Data Hurt" phenomenon. Our method is to compare the confidence intervals of the prediction risk at different sample sizes to confirm the "More Data Hurt" phenomenon. The isotropic case is computational feasible since the limiting spectral distribution of $\Sigma$ is concentrated at one. These explicit forms of confidence intervals are sufficient to demonstrate the “More Data Hurt” phenomenon.
> > >
> > >
> > > 2. *"About convergence speed, ....... , for a converging sequence* $A_n = C+o(1)$, *I would like to have a lower bound on the o(1) term ...... "*
> > >
> > > **Ans:** Thank you for the detailed comment. We are sorry that we have misunderstood your previous comment. We agree with you that Theorem~1 in Bartlett’s paper gives a lower bound of the risk, not the convergence rate. As for the paper ``Benign overfitting in linear regression",  please refer to the latest PNAS version [Link](https://www.pnas.org/content/pnas/early/2020/04/22/1907378117.full.pdf.)
> > >
> > > In our paper, the converging sequence you mentioned $A_n = C + o_p(1)$ can be understood as follows: $A_n$ stands for the prediction risk, $C$ is its limit and $o_p(1)$ is an asymptotically negligible term, which turns out in our paper to have order $1/p$ or $1/\sqrt p$. More specifically, we have shown that either $p(A_n-C)$ or $\sqrt p (A_n-C)$ (corresponds to different types of prediction risk) converges in distribution to a Gaussian random variable with explicit mean and variance. That is $A_n = C + O_p(\frac{1}{p})$ or $A_n = C + O_p(\frac{1}{\sqrt p}).$ According to the normal approximation, we further derive the confidence interval of $A_n$, which further gives both the upper bound and lower bound of $A_n$. Then we can derive the lower bound of the term $o_p(1)$ from the confidence interval of $A_n-C.$

---

### Official Review · AnonReviewer1 · 2020-10-28
**Review of Provable More Data Hurt in High Dimensional Least Squares Estimator**

**Rating:** 6
**Confidence:** 3

**Review:**

This paper investigates the phenomenon of double descent, also referred to as "more data hurts", in high dimensional linear regression using the least square estimator.  In the same setup, previous sharp results were already established in the asymptotic regime. Non-asymptotic results are also known but are less precise. The authors of this paper try to provide a new type of results that fill the gap between the two regimes (asymptotic vs non-asymptotic). To do so they have managed to derive second order (CLT type) asymptotic results for different risks based on more refined random matrix theory results.

#######################################################################
pros:

Asymptotic confidence intervals for different prediction risks  are derived. These results seem new. One of the main applications of the main results is a better explanation of the more data hurts phenomenon. Indeed, using just first order asymptotic results we had to take both n and p to infinity and compare the ratios p/n. Using the finite-sample results in this paper we can now fix p (large enough) and see that, in the overparameterized regime, a larger n leads to a larger prediction risk.

#######################################################################
cons:

While the statistical decomposition of the risk was already known, the novelty of the results in the present paper are only based on known results from random matrix theory which limits its theoretical contribution. Also, Section 4 needs more discussion. As a reader, I was felt abandoned at the end of Section 4, where the it just ended after stating the results. I think at this stage the authors should take the time to explain more their results and how they are different from previous ones. In particular by saying "explicitly" what happens when the sample size grows. Although that was clear from your introduction, but it is always good to draw conclusions for the reader after you state your results and remind your contributions.

#######################################################################
Score:

This paper is well written and the proofs seem sound to me. Overall, I think the present paper is marginally above the acceptance threshold because it seems like an incremental work over previous asymptotic results by using well established CLT results from random matrix theory.

Typos:

* In assumption (B1), did you mean $\lambda_{\min}$ instead of $\lambda$?

---

> ### Author Response · Authors · 2020-11-17
> **Response**
>
> Thank you for the constructive feedback. The paper has been carefully revised according to your comments, and all changes are marked in red. We hope that our point-by-point response below can address your concerns.
>
> Q1. "*The novelty of the results in the present paper are only based on known results from random matrix theory which limits its theoretical contribution.*"
>
> **Ans:** We would like to emphasize the novelty of our work here. Firstly, although first order limits of the prediction risk in high dimensional linear models have already been well studied in the literature, there are very few results on the second-order fluctuations of the prediction due to the technical challenges. While the "*double descent*" curve is a function of the limiting ratio $p/n$, we still need second-order results to characterize the discrepancy between the finite sample distribution of the risk and its first order limit on the curve. Actually, such discrepancy is shown in our paper to be closely related to the model complexity. The CLTs we derived successfully fill this gap.
>
> Secondly, from the technical perspective, indeed we make use of tools from random matrix theory to solve this problem. However nontrivial calculations and familiarity with the most updated theories are required to derive each CLT result for different combinations of model settings. Since we consider random and non-random signals for two types of conditional prediction risk, each scenario leads to a different bias-variance decomposition and hence calls for different tools. In fact, we have not only derived the leading order constants in the limiting mean and variance but also found out smaller order terms to enhance empirical performance for practitioners, all of which justify the novelty and theoretical contributions of this paper.
>
>
> Q2. "*Section 4 needs more discussion. As a reader, I was felt abandoned at the end of Section 4, where it just ended after stating the results. I think at this stage the authors should take the time to explain more their results and how they are different from previous ones. I think at this stage the authors should take the time to explain more their results and how they are different from previous ones. In particular by saying "explicitly" what happens when the sample size grows*"
>
> **Ans:** Thank you for this comment. We have added a new Section 4.4 to summarize our contributions and how our results differ from existing works. Moreover, we have carefully explained the technical limitations in this conclusion section. As for “what happens when the sample size grows”, we have added an example for illustration in the introduction and  provided more details in Appendix F.
>
> Q3. “ *Typos: In assumption (B1), did you mean* $\lambda_{min}$ *instead of* $\lambda$?”
>
> **Ans:** Thank you for the comment. Done. Please see Assumption (B1) in Page 5 of the revised version.

---

### Decision · Program_Chairs · 2021-01-07
**Final Decision**

**Decision:**

Reject

**Comment:**

This paper derives CLT type results for the minimum $\ell_2$ norm least squares estimator allowing both n and p to grow.

Pros:
As one reviewer puts it: Asymptotic confidence intervals for different prediction risks are derived. These results seem new.

Cons:
It's not clear what has been gained by having these results, other than having them.

Reasoning:
Staring at Figure 1 for a while, what jumps out is how little the CI matters. Unless $p\approx n$, the band is essentially uniform around the first-order result derived elsewhere. The claim the authors seem to make at the bottom of page 1 is that, "supposing I have 90 observations and 100 predictors, it may not be so bad to collect 8 more observations. Even though on average I'm worse off, perhaps not for my data?" The flip side of this argument is "why am I using min-norm OLS"? I think that the authors are making the wrong argument in this paper. The point of analyzing this problem is not to understand what happens when $p\approx n$ but to understand why $p \gg n$ is good, and thereby try to justify parameter explosion in deep learning. I should be looking at the left side of Figure 1, not the center. Even the language "more data hurt" is the wrong statement. The point isn't to show that collecting data is bad but to justify adding parameters. We should say "more parameters help". If the authors' proof technique added to the understanding in that case, then this paper would be more convincing. As is, I find it hard to overrule with the reviewers who appear to be mainly on the fence with little enthusiasm.

---

> ### Author Response · Authors · 2021-01-20
> **Response to Final Decision**
>
> We feel sorry and disappointed with this decision.  Hope that the following explanation can help make our contribution clearer and more understandable.
>
> **Response to** "**Cons: ......**"
>
> As we emphasized in the introduction and discussion section, the main contribution of our paper is to provide a fine-grained characterization of the second-order fluctuations of the prediction risks. The “Asymptotic confidence intervals” you mentioned is just a by-product of the central limit theorems we derived. The main reason we dive into this topic is that the existing asymptotic results, which focus on the first-order limit of the prediction risk, cannot fully describe the “more data hurt” phenomenon. There is a non-negligible discrepancy between the prediction risk of a given data sample with its first-order asymptotic limit. What we did is to fully characterize such discrepancy, including how fast the risk converges to its limit and what kind of distribution such discrepancy follows. In this way, we know better about the empirical prediction risk of the given data sample and can give a more precise evaluation of the risk. Furthermore, the theoretical results we derived further justified and explained the “more data hurt” phenomenon.
>
> From a technical point of view, we introduced tools from random matrix theory to derive the second-order asymptotic results, which provides a new perspective for solving technical difficulties. We believe that both the results and technical tools of our paper make unique contributions to the existing over-parameterization literature. This work paves the way for quantifying the randomness of the prediction risk for over-parameterized models.
>
> In the following, we provide some explanations regarding your reasons for the final decision. Hope that it can help eliminate some unnecessary misunderstandings.
>
>
> **Response to** "**Reasoning:......**"
>
> The width of the confidence interval depends on the signal-noise-ratio, signal length $r^2$ and the noise level $\sigma^2$. We provide explicit forms of the CI. Figure 1 here is just one simple example with $r^2=1, \sigma^2=1$. We can easily expand these confidence bands to amplify the impact of CI by altering the model parameters.
>
> Our work is not to describe what happens when $p\approx n$. We assume p and n tend to infinity and $p/n \to c$, here the constant c can be arbitrary numbers. The case $p\gg n$ corresponds to $c\gg 1$, which is within our consideration. No matter how large p is, we can describe its corresponding prediction risk by substituting $c=p/n$ into the established central limit theorems. Please refer to Related Works in Page 3 for more discussions about the case $p\gg n$.
>
> “More Data Hurt” comes from the sample-wise double descent risk curve and refers to the phenomenon that training with more data would hurt the prediction performance of the learned model.  The phenomenon occurs under certain circumstances, especially for some deep learning tasks. [2] experimentally confirms the sample-wise non-monotonicity of the test accuracy on deep neural networks. [3] considers adding one single data point to a linear regression task and analyzes its marginal effect to the test risk. [1] derived the first-order limit of the double descent prediction risk of the min-norm OLS in over-parametrized linear regression. Our paper starts from [1] and at the bottom of Page 1, we use the data example to illustrate the necessity to study second-order fluctuations to compensate for the shortcomings of first-order results. It is better to consider the randomness of the prediction risk and use hypothesis testing to justify the “more data hurt” phenomenon. The target of our paper is not to justify adding parameters. We aim at providing theoretical justification for the "More Data Hurt" phenomenon in the existing literature.
>
> [1] Hastie, T., Montanari, A., Rosset, S., & Tibshirani, R. J. (2019). Surprises in high-dimensional ridgeless least squares interpolation. arXiv preprint arXiv:1903.08560.
>
> [2] Nakkiran, P., Kaplun, G., Bansal, Y., Yang, T., Barak, B., & Sutskever, I. (2019, September). Deep Double Descent: Where Bigger Models and More Data Hurt. In International Conference on Learning Representations.
>
> [3] Nakkiran, Preetum. (2019). More data can hurt for linear regression: Sample-wise double descent. arXiv preprint arXiv:1912.07242,